# Bootstrapping Language-Guided Navigation Learning with Self-Refining Data Flywheel

**Zun Wang**[1,2]* **Jialu Li**[2] **Yicong Hong**[3] **Songze Li**[1] **Kunchang Li**[1] **Shoubin Yu**[2]
**Yi Wang**[1,5] **Yu Qiao**[1] **Yali Wang**[1] **Mohit Bansal**[2] **Limin Wang**[1,4]
[1]Shanghai AI Laboratory  [2]UNC Chapel Hill
[3]Adobe Research  [4]Nanjing University  [5] Shanghai Innovation Institute
{zunwang, jialuli, mbansal}@cs.unc.edu, wanglimin@pjlab.org.cn

## Abstract

Creating high-quality data for training robust language-instructed agents is a long-lasting challenge in embodied AI. In this paper, we introduce a Self-Refining Data Flywheel (SRDF) that generates *high-quality* and *large-scale* navigational instruction-trajectory pairs by iteratively refining the data pool through the collaboration between two models, the instruction generator and the navigator, without any human-in-the-loop annotation. Specifically, SRDF starts with using a base generator to create an initial data pool for training a base navigator, followed by applying the trained navigator to filter the data pool. This leads to higher-fidelity data to train a better generator, which can, in turn, produce higher-quality data for training the next-round navigator. Such a flywheel establishes a data self-refining process, yielding a continuously improved and highly effective dataset for large-scale language-guided navigation learning. Our experiments demonstrate that after several flywheel rounds, the navigator elevates the performance boundary from 70% to 78% SPL on the classic R2R test set, surpassing human performance (76%) for the first time. Meanwhile, this process results in a superior generator, evidenced by a SPICE increase from 23.5 to 26.2, better than all previous VLN instruction generation methods. Finally, we demonstrate the scalability of our method through increasing environment and instruction diversity, and the generalization ability of our pre-trained navigator across various downstream navigation tasks, surpassing state-of-the-art methods by a large margin in all cases.[1]

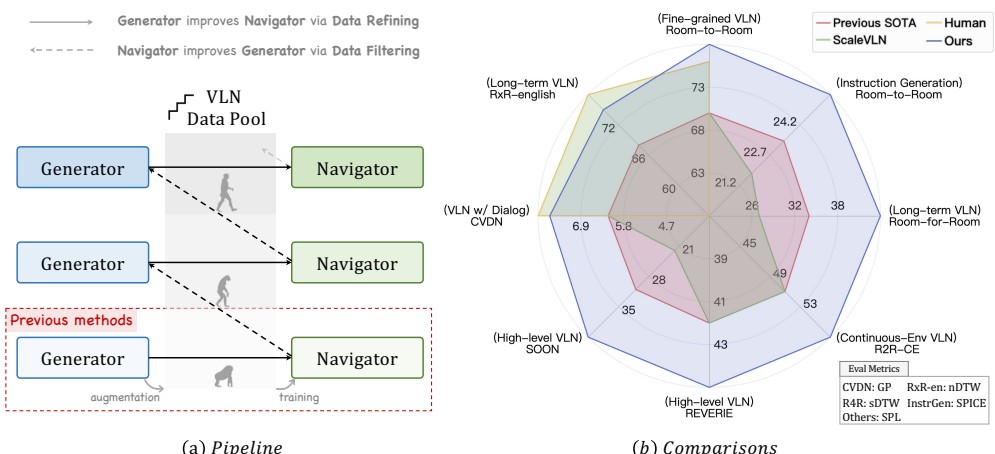

Figure 1: **(a) Our Pipeline:** After using the (instruction) generator to label paths for data augmentation in navigator training, we leverage the trained navigator to filter high-quality data to train a better generator, and the improved generator refines the data pool to train a stronger navigator, iteratively running on the flywheel. **(b) Comparison with state-of-the-art (SoTA) methods**: Our approach significantly outperforms SoTA on all tasks. It also **surpasses human performance** on R2R and approaches human-level results on RxR-English and CVDN (for other tasks, human performance is not reported in their paper). The R2R result is from the test set, while others are from val unseen.

---

[1]Code and data are available at https://github.com/wz0919/VLN-SRDF.
*Interned at Shanghai AI Laboratory.

# 1 INTRODUCTION

The lack of high-quality data is one of the main bottlenecks in training embodied agents to complete real-world human activities. Unlike many other discriminative or generative learning problems, where the data itself naturally formulates a self-supervised learning objective (Devlin, 2018; He et al., 2022) or the data labeling can be facilitated by existing models (Ros et al., 2016; Tian et al., 2024) , training embodied agents usually requires expensive human annotation on complex vision-linguistic contents and physical interactions. In the essential embodied AI problem of Vision-and-Language Navigation (VLN) (Zhang et al., 2024b), one widely considered solution is to first train a trajectory-to-instruction generator (Fried et al., 2018; Tan et al., 2019; Chen et al., 2022c) using a small amount of human-annotated data and then use the generator to caption large-scale trajectories sampled from interactive environments (Hao et al., 2020; Chen et al., 2022c; Wang et al., 2023d). Among them, Hao et al. (2020) demonstrates the effectiveness of scaling the synthetic instruction-trajectory pairs in the existing training environments, while Chen et al. (2022c); Wang et al. (2023d); Li & Bansal (2023) emphasizes the importance of scaling training environments for better generalization ability of agents. However, the quality of the synthetic data, especially the language fidelity, is highly under-investigated. We find that training solely with the large-scale synthetic data ($352\times$ larger in instruction-trajectory pairs, and $13\times$ more diverse in environments) yields worse results compared to training with a small human-annotated dataset (see Table 1), which indicates the need for high-quality instructions beside simply scaling the data and environments.

Though many methods have been proposed for improving data quality in multi-modal understanding and generation tasks (Fang et al., 2023; Li et al., 2024a; Betker et al.; Fang et al., 2024; Li et al., 2024b; Nguyen et al., 2024b), improving synthetic instruction-trajectory pairs for language-guided navigation has several distinct challenges. First, the most straightforward approach is to build a strong instruction generator, but the limited amount of high-quality training data (e.g., R2R (Anderson et al., 2018b) train split has only 14K human-labeled data) makes it challenging to train a robust generator capable of producing high-fidelity instructions for diverse trajectories. Additionally, manually correcting instructions by humans is resource-intensive and costly.

Moreover, the alignment of instruction-trajectory pairs in VLN is hard to evaluate, as the instructions not only contain semantic information (e.g., 'Walk past the table') but also have rich directional information (e.g., 'Turn left') to match with the corresponding trajectory. Besides, the visual elements mentioned in the instructions are also temporally aligned to panoramas in multiple scenes. As a result, traditional metrics like CLIP score (Radford et al., 2021) struggle to evaluate such multi-scene directional and semantic alignment, as they only capture single-scene semantic relationships.

Table 1: Performance (on R2R validation unseen split) on different datasets solely. Directly training with R2R yields the best SPL compared to training with other augmentation datasets.

| Training Data | #data | #Env. | SR↑ | SPL↑ |
|---|---|---|---|---|
| R2R | 14K | 61 | 65.9 | **55.9** |
| Prevalent | 1.0M | 60 | **67.1** | 54.8 |
| ScaleVLN | 4.9M | 800 | 63.9 | 50.1 |

In this work, we propose a Self-Refining Data Flywheel, SRDF, that automatically evaluates and improves the quality of the generated instructions at scale, through an iterative collaboration between the navigator and the instruction generator. As shown in Figure 1 (a), our first step is similar to ScaleVLN's process (Wang et al., 2023d), which trains an instruction generator using the original human-labeled data, then generates instructions for unlabeled paths sampled from 800 HM3D training environments and trains a strong navigator using the generated data. Then, for evaluating the generated instructions, we propose to use the trained navigator's path-fidelity score (nDTW (Ilharco et al., 2019) and SPL (Anderson et al., 2018a), measuring how closely the navigator's followed path matches the original trajectory) as the similarity score. Since the trained navigator is strong enough (achieves human-level performance in Wang et al. (2023d)) and has already learned to connect visual landmarks and directional cues to actions effectively, its fidelity in following the instructions can naturally reflect how well the instruction-trajectory pairs are aligned. It also avoids manually setting vague thresholds when using CLIP-score-like metrics for filtering. In our case, SPL=1 yields a perfect trajectory match. After using the navigator as a filter to obtain a high-quality subset of the generated data, this subset of data will be used to train a better instruction generator in the next iteration. The improved instruction generator re-generates instructions for bad samples to produce better datasets, which is used to train a stronger navigator. The process iterates, with the navigator improving the instruction generator via data filtering, and the instruction generator improving the

navigator via data refining, ultimately producing both a highly capable navigator and instruction generator, along with a substantially high-quality synthetic VLN dataset.

We build our flywheel upon the R2R dataset (Anderson et al., 2018b) and provide detailed analysis. Empirically, we show that our navigator and instruction generator can iteratively improve each other with our SRDF. We also demonstrate the scalability of our method: the instruction generator consistently improves with additional environments and data, and the navigator benefits more from increased instruction diversity when trained with our high-quality instructions compared to with low-quality datasets. On the R2R dataset, our approach surpasses previous state-of-the-art results by a wide margin in both instruction following and generation, and notably, for the first time, we significantly surpass human performance (76% SPL) in instruction following, demonstrating the effectiveness of our SRDF to improve data quality. Furthermore, we evaluate the transferability of our pre-trained navigator across various downstream navigation tasks, including VLN with dialogue-based instructions (CVDN (Thomason et al., 2020)), long-term VLN (RxR-English (Ku et al., 2020), R4R (Zhu et al., 2020)), high-level VLN (SOON (Zhu et al., 2021), REVERIE (Qi et al., 2020)), and even VLN in continuous environments (R2R-CE (Krantz et al., 2020)). As shown in Figure 1 (b), we achieve state-of-the-art performance on all the tasks, while approaching human performance for RxR-English and CVDN, underscoring the superior quality of our generated data and the robust transferability of our pre-trained navigator.

## 2 RELATED WORK

**Vision-and-Language Navigation.** VLN requires an agent to navigate in unseen environments based on natural language instructions. Numerous datasets have been proposed for this task (Anderson et al., 2018b; Ku et al., 2020; Qi et al., 2020; Shridhar et al., 2020; Thomason et al., 2020; Padmakumar et al., 2022; Nguyen & Daumé III, 2019; Chen et al., 2019; Kim et al., 2021), spanning both indoor and outdoor environments with varied levels of instruction detail. The limited availability of human-annotated data for training generalizable VLN agents to achieve near-human performance is a key challenge due to the high cost of collecting instruction-trajectory pairs. To address this, various data augmentation approaches have been explored. Some focus on scaling environments by editing existing ones (Li et al., 2022; Liu et al., 2021b) or generating new ones with text-to-image models (Li & Bansal, 2023). Others scale instruction data by training instruction generators to generate instructions for unannotated paths (Hao et al., 2020; Zhang & Kordjamshidi, 2023; Zhang et al., 2024a), or by leveraging large sets of rendered environments from simulators (e.g., HM3D (Ramakrishnan et al., 2021), Gibson (Xia et al., 2018)) (Wang et al., 2023d; Kamath et al., 2022; Chen et al., 2022c). While data scaling has been effective for VLN, the quality of the data, particularly the alignment between instructions and trajectories, remains under-explored. In this paper, we investigate the impact of data quality on VLN and propose a data flywheel that iteratively refines itself through collaboration between the navigator and the generator.

**High-Quality Multimodal Dataset Curation.** Many multimodal studies show that improving data quality can significantly enhance model performance, either through advanced dataset filtering (Fang et al., 2023; Li et al., 2024a; Gadre et al., 2024; Sun et al., 2023) or refining captions with strong models (Betker et al.; Fang et al., 2024; Li et al., 2024b; Nguyen et al., 2024b; Wang et al., 2024c; Tan et al., 2024). Recently, Segment Anything (SAM) (Kirillov et al., 2023) demonstrated how data and models can improve each other through a data flywheel with a human-in-the-loop process, evolving from model-assisted to fully automated annotation. Our data flywheel similarly integrates filtering and re-captioning data via navigator verification and generator refinement, operating in a data-model loop like SAM, but without any human intervention.

**Self-Improving Language Models.** Studies show that Large Language Models (LLMs) can improve themselves by training on their own generated outputs across tasks like programming (Haluptzok et al., 2022), summarization (Patil et al., 2024), question-answering (Lee et al., 2024; Yu et al., 2024), reasoning (Prasad et al., 2024), and others (Li et al., 2023a; Madaan et al., 2024; Zhou et al., 2024c), where the quality of the self-generated data is ensured via human (Ouyang et al., 2022; Bai et al., 2022a), off-the-shelf verifiers/reward models (Ni et al., 2023; Wang et al., 2019; Dou et al., 2024; Bai et al., 2022b; Lee et al., 2023; Nguyen et al., 2024a) or model's self feedback (Yuan et al., 2024; Wu et al., 2024; Wang et al., 2024d). In our pipeline, the instruction generator iteratively self-improves using its own generated data. Unlike prior work, our approach establishes a fully automated, multi-round, two-model mutual improvement process, enabling the navigator, the generator, and the dataset to evolve concurrently through continuous model-driven feedback.

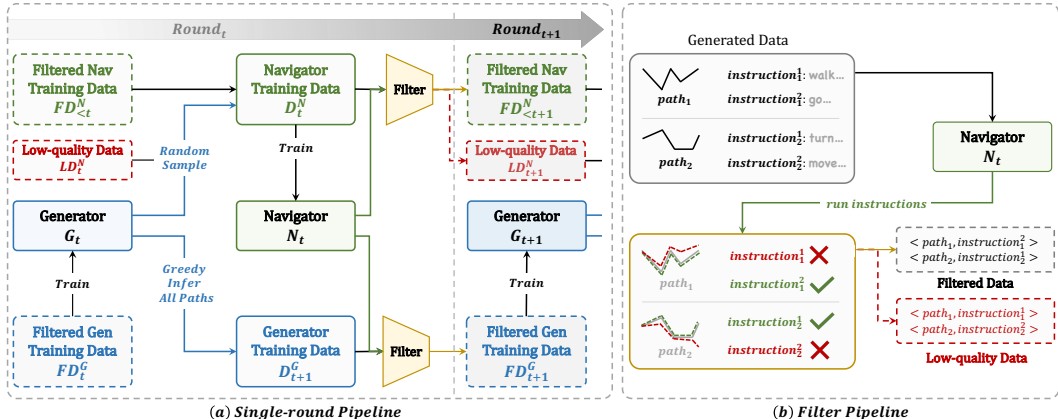

Figure 2: (a) Our (instruction) generator refines low-quality data filtered in the previous round via sampling to train the next-round navigator, and greedily generates high-confidence instructions as candidates to train next-round generator. Then the navigator filters the data for further use. (b) In filtering, the navigator re-runs instructions to compute the path-path similarity score (nDTW & SPL). High-fidelity data is kept for training while low-quality data will be refined.

## 3 METHODOLOGY

### 3.1 BACKGROUND

VLN data typically consists of instruction-trajectory pairs, where a trajectory represents a path in a 3D environment, and the corresponding instruction guides the agent to follow it. This data can be used for training either instruction-to-action navigators or trajectory-to-instruction generators. Since manually annotating trajectories is costly, a common approach is to first train a generator on limited human-annotated data and then use it to generate instructions for paths in unlabeled environments, which are subsequently used to augment navigator training.

While this method helps increase the data amount, it poses challenges regarding the quality and fidelity of the generated instructions. These challenges imply two essential problems – how to generate better data and evaluate the data quality, which we aim to address in the following sections.

### 3.2 SRDF: SELF-REFINING DATA FLYWHEEL

In this section, we introduce the Self-Refining Data Flywheel (SRDF) to tackle the challenge of evaluating and improving VLN data to bootstrap its learning. Broadly, our system comprises a navigator, $N$, and an instruction generator, $G$, shown in Figure 1 (a). We use $N$ to assist $G$ by optimizing G using the data filtered by $N$, while $G$ enhances $N$ by refining the low-quality data. This iterative process is repeated multiple times, consistently enhancing both $G$ and $N$'s performance.

**Main Components.** Our SRDF requires the following resources at the beginning: (1) seed data $D_{Seed}$, typically human-annotated, for training a base $G$ and $N$ (2) unlabeled trajectory pool, $D_{Traj}$, usually collected from large-scale environment datasets for generating new training data.

**Training Base Instruction Generator.** Most previous works (Fried et al., 2018; Tan et al., 2019; Dou & Peng, 2022) train instruction generators from scratch, which limits their ability to generalize text effectively. Some recent approaches leverage pretrained vision-and-language models but neglect critical directional information during trajectory encoding (Li & Bansal, 2023; Kong et al., 2024), leading to instructions with inconsistent or incorrect directional cues.

We hypothesize that an effective instruction generator should be capable of understanding both multi-image inputs and interleaved image-text inputs. Multi-image understanding is crucial for accurately encoding multi-panorama trajectories and interleaved image-text comprehension helps encode directional images within the raw text space, enabling simple yet effective visual prompting of trajectories. To achieve this, we utilize Mantis (Jiang et al., 2024), an interleaved multi-image mul-

timodal large language model (MLLM), which includes a SigLIP vision encoder (Zhai et al., 2023), a multimodal projector, and a LLaMA-3 language model backbone (Dubey et al., 2024), pretrained on a curated corpus of interleaved image-text pairs. We use our designed template (see Appendix 3, 13 for details) to convert the instruction-trajectory pairs from $D_{Seed}$ into supervised fine-tuning (SFT) data, and then LoRA-fine-tune the language backbone using this converted data. This results in a robust instruction generator $G_1$, which serves as the foundation for our data generation process.

**Generating Base Training Data.** Once the base instruction generator, $G_1$, is trained, it is used to generate two types of data from the unlabeled trajectories, $D_{Traj}$: one for training the navigator and another for improving the round-2 instruction generator. For data training the instruction generator, we use greedy decoding to generate the most reliable instruction for each trajectory to form $D_2^G$. For data used to train the navigator, we prioritize diversity by generating multiple instructions via random sampling to form $D_1^N$. The data generation step aims to provide sufficient training data for both improving the navigator and the instruction generator in subsequent iterations.

**Training Base Navigator.** We employ the DUET model (Chen et al., 2022b) as our navigator. The model is pre-trained using $D_1^N$ and subsequently fine-tuned on $D_{Seed}$. This results in a highly capable navigator $N_1$, which will be used to evaluate the quality of the generated data.

**Evaluating and Filtering the Generated Data.** We propose using the trained navigator $N_1$ to self-evaluate the generated data $D_1^N$. Intuitively, if the navigator successfully follows the generated instruction and navigates along the original path, it suggests that the instruction is well-aligned with the trajectory and the strong performance of this navigator (Could achieve near-human performance in previous work (Wang et al., 2023d)) further ensures its reliability. Such self-consistency transforms the challenging task of computing instruction-trajectory similarity into the simpler problem of computing trajectory-trajectory similarity.

We run the navigator on the generated instructions and compute trajectory-trajectory similarity scores. We filter high-quality data $FD_2^G$ for the round-2 instruction generator training by selecting instances from $D_2^G$ with SPL=1 (indicating the navigator perfectly follows the path). For $D_1^N$ (navigator training data), we filter with nDTW$\geqslant$0.9 (indicating very close alignment between trajectories) to get $FD_{<2}^N$, to use in the round-2 training, as shown in Figure 2. This filtered-out high-quality data will be kept in the follow-up round, and we only re-generate low-quality data $LD_2^N$ filtered by nDTW<0.9. This filtering step ensures that only reliable data is used for subsequent iterations of training, reducing data noise and increasing alignment quality.

**Iterative Self-Refining.** The core of SRDF is its iterative loop between the instruction generator and the navigator, where each model contributes to improving the other by providing data feedback. Specifically, at each iteration $t$, we first train the generator $G_t$ using the filtered high-quality data $FD_t^G$. Then we use $G_t$ to generate new navigation-training data, $ND_t^N$, for previously bad samples, $LD_t^N$, and new generator training data, $D_{t+1}^G$, for $D_{Traj}$, following the same details in generating base training data. Then we Combine $ND_t^N$ with previously filtered navigation data, $FD_{<t}^N$, to form the complete dataset, $D_t^N$, for training the navigator. Note that the whole data size won't change as we are only refining base samples. After training the navigator $N_t$ using $D_t^N$, we use $N_t$ to filter high-quality data subsets, resulting in $FD_{t+1}^G$ filtered from $D_{t+1}^G$ and $FND_t^N$ from $ND_t^N$. Finally, we combine the newly filtered navigation data, $FND_t^N$, to form $FD_{<t+1}^N$ for the next-round training. This looping mechanism ensures that the data quality continually improves through a self-refining process. Each iteration benefits from the enhanced quality of both the instruction generator and the navigator, ultimately yielding a high-quality dataset and highly capable models. To summarize, the pseudocodes of the SRDF are detailed in Appendix Alg. 1.

### 3.3 FINAL DATASET

We build our flywheel upon the R2R dataset (Anderson et al., 2018b) as $D_{Seed}$, containing 14,039 human-annotated training data, along with the 178,270 and 2,890,267 unlabelled trajectories from MP3D (Chang et al., 2017) and HM3D (Wang et al., 2023d) environments, respectively, as $D_{Traj}$. We run the flywheel for three rounds to create the final dataset, named SRDF-20M. This dataset consists of 19.5 million pre-training examples $D_3^N$, with 6 instructions generated for each HM3D path

Table 2: Statistics of training data on different VLN datasets.

| Dataset | Instruction | #Env. | #Instr. | #Vocab. | Instr. Length |
|---|---|---|---|---|---|
| R2R (Anderson et al., 2018b) | | 61 | 14,039 | 3,063 | 26.33 |
| RxR-en (Ku et al., 2020) | | 60 | 26,464 | 7,249 | 102.13 |
| REVERIE (Qi et al., 2020) | Manually | 60 | 10,466 | 1,140 | 18.64 |
| CVDN (Thomason et al., 2020) | Labelled | 57 | 4,742 | 2,068 | 53.21 |
| SOON (Zhu et al., 2021) | | 34 | 2,780 | 735 | 44.09 |
| R4R (Zhu et al., 2020) | | 59 | 233,532 | 3,004 | 52.25 |
| Prevalent (Hao et al., 2020) | | 60 | 1,069,620 | 993 | 24.23 |
| Marky (Wang et al., 2022b) | | 60 | 333,777 | 2,231 | 99.45 |
| AutoVLN (Chen et al., 2022c) | Generated | 900 | 217,703 | 1,696 | 20.52 |
| ScaleVLN (Wang et al., 2023d) | | 1289 | 4,941,710 | 172 | 21.61 |
| SRDF-20M (Ours) | | 860 | 20,417,874 | 10,363 | 24.05 |

(via top-3 sampling) and 12 instructions (6 top-3 and 6 top-5 sampling) for each MP3D path, and 0.9M greedy-decoded filtered data $FD_4^G$ (which will be used to fine-tune the pretrained navigator and the third-round generator here). Overall, we generated 20M instructions across 860 environments for navigator training.

**Comparison to Previous VLN Datasets.** Table 2 presents detailed statistics of our dataset and previous VLN datasets. A common issue in existing augmentation datasets is the lack of vocabulary diversity. For instance, prior datasets like Prevalent (Hao et al., 2020), despite being significantly larger than R2R, possess only 1/3 of R2R's vocabulary size, even with 76 times more instructions. This issue is even more pronounced in ScaleVLN (Wang et al., 2023d), which suffers from a highly limited vocabulary. In contrast, our dataset contains over 10,000 unique words and 20 million instructions— three times the vocabulary diversity and 1,454 times the instruction count compared to the original R2R dataset — surpassing all previous human-labeled and synthetic VLN datasets in both vocabulary richness and instruction quantity. Importantly, this substantial increase in size and diversity is achieved without compromising quality, due to the robustness of our instruction generator and the high-precision filtering process guided by our strong navigator. We also provide some visualization examples of our generated instructions in Appendix Figure 4 and 5.

## 4 EXPERIMENTS

### 4.1 EXPERIMENTAL SETUP

**Datasets and Evaluation Metrics.** We establish our data flywheel and perform ablation studies primarily on the R2R dataset, while also assessing the transferability of our pre-trained navigator on a range of downstream tasks. These include fine-grained VLN (R2R, RxR-English), VLN with dialog history (CVDN), high-level VLN (REVERIE, SOON), long-term VLN (R4R), and VLN in continuous environments (R2R-CE). Each dataset is split into training, val_seen, and val_unseen sets, with R2R, CVDN, REVERIE, SOON, and R2R-CE also containing test splits. The statistics for the training splits are summarized in Table 2 (manually-labeled datasets), and further details can be found in the Appendix.

We evaluate our navigator using the standard path-fidelity metrics including Success Rate (SR), Success Rate Weighted by Path Length (SPL) (Anderson et al., 2018a), Goal Progress (GP) (Thomason et al., 2020), Navigation Error (NE), normalized Dynamic Time Warping (nDTW) (Ilharco et al., 2019) and Success Rate Weighted by Dynamic Time Warping (sDTW) (Ilharco et al., 2019). We leave the details of these metrics to Appendix.

Although REVERIE and SOON include object grounding tasks, we focus on evaluating navigation performance, as our generated dataset is specifically designed to enhance navigation tasks. We use SPL as the primary metric for R2R, REVERIE, SOON, and R2R-CE, nDTW for RxR-English, sDTW for R4R, and GP for CVDN.

For evaluating our instruction generator, we assess the linguistic quality of the generated instructions using standard text similarity metrics such as BLEU (Papineni et al., 2002), Meteor (Banerjee & Lavie, 2005), and Rouge (Lin, 2004), along with commonly used image captioning metrics, includ-

Table 3: Navigator and instruction generator results in different rounds in R2R val unseen split.

| Method | Instruction Following | | | | Instruction Generation | | | | | | |
|---|---|---|---|---|---|---|---|---|---|---|---|
| | NE↓ | OSR↑ | SR↑ | SPL↑ | SPICE↑ | SPICE-D↑ | Bleu-1↑ | Bleu-4↑ | CIDEr↑ | Meteor↑ | Rouge↑ |
| Baseline | 2.37 | 85.5 | 78.6 | 69.9 | 21.8 | 28.0 | 72.5 | 27.7 | 42.2 | 23.6 | 49.0 |
| Ours (round 1) | 1.95 | 87.1 | 82.4 | 75.9 | 23.7 | 28.4 | 71.4 | 29.5 | 46.5 | 23.1 | 50.2 |
| Ours (round 2) | 1.81 | 88.5 | 83.6 | 77.3 | 25.2 | 29.9 | 73.7 | **31.0** | **50.7** | 24.2 | **51.3** |
| Ours (round 3) | **1.76** | **89.6** | **84.4** | **77.6** | **25.7** | **30.4** | **74.5** | 30.8 | 49.7 | **24.5** | **51.3** |

ing SPICE (Anderson et al., 2016) and CIDEr (Vedantam et al., 2015). Additionally, we consider SPICE-D (Zeng et al., 2023), a directional-aware variant of SPICE tailored for VLN instruction evaluation. We adopt SPICE as our primary metric.

**Implementation.**   We utilize InternViT (Chen et al., 2024), a powerful ViT with 6B parameters, as the visual encoder in our instruction-following experiments unless otherwise specified. For continuous environments, we employ CLIP-B/16 (Radford et al., 2021) as the visual backbone.

In our data flywheel, we pre-train the DUET navigator from scratch for 45,000 iterations using a batch size of 1024 and a learning rate of $5 \times 10^{-5}$ on 8 NVIDIA Tesla A100 GPUs. Multiple checkpoints are fine-tuned to select the best pre-training model. The selected model is then fine-tuned for 6K iterations with a batch size of 16 on a single GPU using only the R2R dataset. For the instruction generator, we fine-tune Mantis-8B-SigLIP (Jiang et al., 2024) using LoRA, applying it to the query and value layers in each transformer block. Initially, both the navigator and instruction generator are trained from random weights, while subsequent rounds use previous-round weights.

After final-round navigator pre-training, we fine-tune the navigator for various downstream tasks, using PanoGen (Li & Bansal, 2023) for MP3D-level environment augmentation. For augmentation, we use round-3 filtered data $FD_4^G$ for R2R, ScaleVLN(REVERIE) (Wang et al., 2023d) for REVERIE, and marky-English (Wang et al., 2022b) for RxR-English, and no augmentation for other tasks. For R2R-CE, we fine-tune our pre-trained navigator on ETPNav (An et al., 2023b) with a waypoint predictor (Hong et al., 2022; An et al., 2022). Modules not pre-trained, such as the object grounding module in REVERIE and the depth-image embedding module in ETPNav, are randomly initialized and trained from scratch. We also use $FD_4^G$ to fine-tune the 3rd-round generator to build the final instruction generator. This process can also be viewed as the fourth round of generator training.

## 4.2   Flywheel Running Results

Table 3 presents the results for both the instruction generator and navigator across all rounds. We follow ScaleVLN (Wang et al., 2023d) to train the navigator with ScaleVLN-HM3D and Prevalent data for augmentation, while we use InternViT (Chen et al., 2024) features for fair comparison. The instruction generator for ScaleVLN is EnvDrop (Tan et al., 2019). In round 1, our new instruction generator, fine-tuned on R2R with LoRA using the pre-trained Mantis, significantly surpasses EnvDrop. This results in a substantial SR boost for the navigator from 78.6% to 82.4%.

**Navigator and Instruction Generator Improve Each Other.**   At each round, we use the navigator to filter high-quality data $FD_t^G$ to re-train the instruction generator, and use the improved instruction generator to refine low-quality instructions $LD_t^N$ to re-train the navigator. Despite the strong performance of the round 1 baseline, the generator is further improved by incorporating navigator-filtered data in round 2. The high-quality data filtered by the navigator leads to +1.5 SPICE and +4.2 CIDEr. This trend continues in round 3, where SPICE reaches 25.7, while other metrics remain stable, demonstrating the crucial role of the navigator in enhancing the instruction generator via data filtering. For the navigator, the data-refining process leads to continuous improvements in navigation performance with +1.2% SR in round 2, and +0.8% SR in round 3, underscoring the importance of the generator data refinement in enhancing the navigator, as well as the effectiveness of iterative navigator-generator collaboration to build an effective self-refining flywheel.

## 4.3   Analysis

**Comparison of Different Scoring Functions.**   We analyzed classical filtering methods will likely fail to capture complex path-instruction similarity in the previous discussion. In Table 4, we further verify the importance of our navigator-filtering compared to other filtering baselines. Using our

Table 5: Results of instruction diversity (#instr. per path) in navigator training in val unseen split.

| Aug Data | NE↓ | SR↑ | SPL↑ |
|---|---|---|---|
| Prev $_{\#Instr=1}$ | 3.21 | 71.86 | 61.04 |
| Ours $_{\#Instr=1}$ | **2.97** | **73.86** | **63.58** |
| Prev $_{\#Instr=3}$ | 3.12 | 72.67 | 62.53 |
| Ours $_{\#Instr=3}$ | **2.81** | **75.21** | **64.56** |
| Prev $_{\#Instr=6}$ | 3.07 | 72.84 | 63.12 |
| Ours $_{\#Instr=6}$ | **2.55** | **76.93** | **66.89** |
| Ours $_{\#Instr=12}$ | 2.59 | 77.05 | 66.53 |

Table 6: Results of different additional augmentation data in instruction generator training in val unseen split.

| Additional Data | SPICE↑ | Bleu-4↑ | CIDEr↑ | Rouge↑ |
|---|---|---|---|---|
| - | 23.7 | 29.5 | 46.5 | 50.2 |
| ScaleVLN Data | 23.5 | 29.0 | 46.1 | 49.8 |
| Prevalent Data | 23.6 | 29.3 | 46.7 | 50.1 |
| 100-HM3D-Env Ours | 23.9 | 29.8 | 47.8 | 50.0 |
| 200-HM3D-Env Ours | 24.2 | 30.1 | 49.1 | 50.3 |
| 400-HM3D-Env Ours | 24.6 | 30.3 | 48.9 | 50.7 |
| 800-HM3D-Env Ours | **25.2** | **31.0** | **50.7** | **51.3** |
| 800-HM3D-Env Ours (Sample) | 24.8 | 30.3 | 48.3 | 51.0 |

round 1 instruction generator, we produce instructions for 783 trajectories from the validation unseen split, rank them based on various scoring functions, and filter the top 400 to assess their similarity to GT instructions. 'No filter' refers to the average score of all 783 instructions, and intuitively, more similar instructions should yield higher NLP metric scores.

We compare our navigator's nDTW-score with CLIP-Sim (Radford et al., 2021) and Mantis score (Jiang et al., 2024). CLIP-Sim is computed by averaging image-instruction similarities across all observations in the trajectory, while Mantis-score is produced by inputting the path as interleaved image-text pairs and asking Mantis to provide a similarity score. Results show that the Mantis score fails to improve over the baseline, likely due to the trajectories is too complex to understand for the MLLM. CLIP-Sim provides a slight SPICE improvement, possibly because it can capture some landmark-level similarities between the trajectory images and the instructions, but does not improve SPICE-D as it lacks directional understanding. In contrast, our Navigator-nDTW similarity filtering method successfully identifies high-quality instructions, leading to a substantial improvement, demonstrating our navigator-nDTW captures path-instruction similarities much better than others.

**Effect of Instruction Diversity in Navigator Training**
We assess the impact of instruction diversity by training the navigator with different numbers of instructions per path ($\#instr = 1, 3, 6, 12$) on the MP3D environments, as shown in Table 5. We use CLIP-B/16 as the visual feature to establish a well-known baseline (Li & Bansal, 2023; Wang et al., 2023d) with the Prevalent dataset, while "Our" uses instructions generated by our round 2 instruction generator. Compared to Prevalent, our instructions consistently

Table 4: Effects of Scoring Functions.

| Filter | SPICE↑ | SPICE-D↑ | CIDEr↑ |
|---|---|---|---|
| No Filter | 23.7 | 28.4 | 46.5 |
| CLIP-Sim | 24.4 | 28.7 | 45.8 |
| Mantis-Score | 23.6 | 28.2 | 48.3 |
| Navigator-nDTW | **25.4** | **30.6** | **53.9** |

achieve stronger downstream results at each $\#instr$ level, emphasizing the importance of instruction quality. Our navigator also benefits significantly when increasing $\#instr$ from 1 to 3 and 3 to 6, while Prevalent's performance saturates after 3, demonstrating that scaling instruction diversity will be more effective when instruction quality is higher. Increasing $\#instr$ to 12 yields similar results to 6, suggesting that $\#instr = 6$ is an optimal balance for training.

**Effect of Additional Data in Instruction Generator Training.** In Table 6, we examine the importance of high-quality data in instruction generator training, and the potential scalability of our pipelines by evaluating the influence of environment numbers. The round-1 generator without supplementary data serves as the baseline. Adding the ScaleVLN dataset does not improve performance, likely due to its low diversity, which limits generalization in text generation tasks.

When training with the Prevalent dataset, which has greater diversity, performance gains remain minimal, possibly because of the data noise, as it also shows low quality in Table 1. In contrast, adding data from ours with increased environments (we split $FD_2^G$ by environments) consistently enhances performance. This improvement is likely due to both the diversity and quality of our data, which are carefully maintained throughout the process, boosting the SPICE from 23.7 to 25.2.

We also experimented with training using sampled versus greedy-decoded instructions. Sampled instructions resulted in slightly lower performance, suggesting they may introduce noise, whereas greedy-decoded instructions, produced with higher confidence, are more reliable. These results show the strong extensibility of our pipeline, and the critical role of high-quality data, both in instruction generator training.

Table 7: Comparison of single-run performance on R2R and R2R-CE datasets. Best results are marked in **bold blue** and second best in **bold**.

| Methods | Room-to-Room Dataset | | | | | | | | Room-to-Room-CE Dataset | | | | | |
| | Validation-Unseen | | | | Test-Unseen | | | | Validation-Unseen | | | Test-Unseen | | |
| | NE↓ | OSR↑ | SR↑ | SPL↑ | NE↓ | OSR↑ | SR↑ | SPL↑ | NE↓ | SR↑ | SPL↑ | NE↓ | SR↑ | SPL↑ |
|---|---|---|---|---|---|---|---|---|---|---|---|---|---|---|
| Human | - | - | - | - | 1.61 | 90 | 86 | 76 | - | - | - | - | - | - |
| Seq2Seq (Anderson et al., 2018b) | 7.81 | 28 | 21 | - | 7.85 | 27 | 20 | - | - | - | - | - | - | - |
| Speaker Follower (Fried et al., 2018) | 6.62 | 45 | 36 | - | 6.62 | - | 35 | 28 | - | - | - | - | - | - |
| EnvDrop (Tan et al., 2019) | 5.22 | - | 52 | 48 | 5.23 | 59 | 51 | 47 | - | - | - | - | - | - |
| VLN↻BERT (Hong et al., 2021) | 3.93 | - | 63 | 57 | 4.09 | 70 | 63 | 57 | - | - | - | - | - | - |
| HAMT (Chen et al., 2021) | 2.29 | - | 66 | 61 | 3.93 | 72 | 65 | 60 | - | - | - | - | - | - |
| HOP (Qiao et al., 2022) | 3.80 | - | 64 | 57 | 3.83 | - | 64 | 59 | - | - | - | - | - | - |
| DUET (Chen et al., 2022b) | 3.31 | 81 | 72 | 60 | 3.65 | 76 | 69 | 59 | - | - | - | - | - | - |
| BEVBert (An et al., 2023a) | 2.81 | 84 | 75 | 64 | 3.13 | 81 | 73 | 62 | 4.57 | 59 | 50 | **4.70** | **59** | **50** |
| ScaleVLN (Wang et al., 2023d) | **2.09** | **88** | **81** | **70** | **2.27** | **86** | **80** | **70** | 4.80 | 55 | 51 | 5.11 | 55 | 50 |
| GridMM (Wang et al., 2023c) | 2.83 | - | 75 | 64 | 3.13 | - | 73 | 62 | 5.11 | 49 | 41 | 5.64 | 46 | 39 |
| ETPNav (An et al., 2023b) | - | - | - | - | - | - | - | - | 4.71 | 57 | 49 | 5.12 | 55 | 48 |
| HNR (Wang et al., 2024e) | - | - | - | - | - | - | - | - | **4.42** | **61** | **51** | 4.81 | 58 | 50 |
| NaviLLM (Zheng et al., 2024) | 3.51 | - | 67 | 59 | 3.71 | - | 68 | 60 | - | - | - | - | - | - |
| NavGPT-2 (Zhou et al., 2024a) | 2.84 | 84 | 74 | 61 | 3.33 | 80 | 72 | 60 | - | - | - | - | - | - |
| VER (Liu et al., 2024) | 2.80 | - | 76 | 65 | 2.74 | - | 76 | 66 | - | - | - | - | - | - |
| MAGIC-L (Wang et al., 2024b) | 2.22 | 86 | 79 | **70** | 2.75 | 82 | 77 | 69 | - | - | - | - | - | - |
| GOAT (Wang et al., 2024a) | 2.40 | 85 | 78 | 68 | 3.04 | 80 | 75 | 65 | - | - | - | - | - | - |
| SRDF (Ours) | **1.62** | **90** | **86** | **79** | **1.82** | **89** | **85** | **78** | **4.12** | **65** | **57** | **4.35** | **64** | **56** |

Table 8: Comparison with previous methods on various downstream tasks. † indicates the RxR-en results are reproduced using their officially released checkpoints. ⋆ means pre-exploration methods. Best results are marked in **bold blue** and second best in **bold**.

| Methods | RxR-english | | R4R | | CVDN | | REVERIE | | | | SOON | | | |
| | Val unseen | | Val unseen | | Val | Test | Val unseen | | Test unseen | | Val unseen | | Test unseen | |
| | SR↑ | nDTW↑ | SR↑ | sDTW↑ | GP↑ | GP↑ | SR↑ | SPL↑ | SR↑ | SPL↑ | SR↑ | SPL↑ | SR↑ | SPL↑ |
|---|---|---|---|---|---|---|---|---|---|---|---|---|---|---|
| HAMT† (Chen et al., 2021) | 56.4 | 63.0 | 44.6 | 31.8 | 5.13 | 5.58 | 33.0 | 30.2 | 30.4 | 26.7 | - | - | - | - |
| MARVAL (Kamath et al., 2022) | 64.7 | 70.4 | - | - | - | - | - | - | - | - | - | - | - | - |
| DUET (Chen et al., 2022b) | - | - | - | - | - | - | 47.0 | 33.7 | 52.5 | 36.1 | 36.3 | 22.6 | 33.4 | 21.4 |
| AutoVLN (Chen et al., 2022c) | - | - | - | - | - | - | 55.9 | 40.9 | 55.2 | 38.9 | **41.0** | **30.7** | 40.4 | **27.9** |
| RREx-Bot⋆ (Sigurdsson et al., 2023) | - | - | - | - | - | - | 61.0 | 58.8 | 65.9 | 62.0 | 49.2 | 48.6 | 47.5 | 47.1 |
| BEVBert† (An et al., 2023a) | 66.7 | 69.6 | - | - | - | - | 51.8 | 36.4 | 52.8 | 36.4 | - | - | - | - |
| ScaleVLN (Wang et al., 2023d) | - | - | - | - | 6.12 | 6.97 | 57.0 | 41.9 | 56.1 | 39.5 | - | - | - | - |
| NaviLLM (Zheng et al., 2024) | - | - | - | - | **6.16** | **7.90** | 44.6 | 36.6 | 43.5 | 34.5 | 38.3 | 29.2 | 35.0 | 26.2 |
| VER (Liu et al., 2024) | - | - | 47.0 | 33.0 | - | - | 56.0 | 39.7 | 56.8 | 38.8 | - | - | - | - |
| PRET† (Lu et al., 2024) | 71.0 | **70.9** | - | - | - | - | - | - | - | - | - | - | - | - |
| MAGIC-L (Wang et al., 2024b) | **72.9** | 68.1 | - | - | - | - | - | - | - | - | - | - | - | - |
| VLN-Copilot (Qiao et al., 2024) | - | - | - | - | - | - | **57.4** | **43.6** | **57.8** | **42.3** | - | - | - | - |
| GOAT (Wang et al., 2024a) | 68.2 | 66.8 | - | - | - | - | 53.4 | 36.7 | 57.7 | 40.5 | 40.4 | 28.1 | **40.5** | 25.2 |
| SRDF (Ours) | **78.8** | **74.4** | **64.4** | **44.6** | **7.67** | **8.19** | **60.4** | **45.4** | **61.4** | **47.7** | **50.3** | **41.7** | **46.6** | **37.9** |

## 4.4 COMPARISION WITH STATE OF THE ARTS

**R2R and R2R-CE.** Table 7 compares agent performance on the R2R and R2R-CE datasets. The DUET navigator, trained on our high-quality datasets, improves SoTA SPL (ScaleVLN (Wang et al., 2023d)) by 8% on the R2R test set, demonstrating our data's strong instruction-trajectory alignment that allows effective decision-making learning. Additionally, The gap between oracle success and success is reduced to 4%, compared to the previous best of 6%, highlighting that our data provides stronger clues for the agent to learn when to stop. Notably, our data-centric approach yields greater improvements compared to most model design modifications, highlighting the importance of building high-quality data to boost model performance. For R2R-CE, despite ETPNav using in-domain pre-training with Habitat-rendered RGBD images (Savva et al., 2019), our model, using no-rendered images from MP3D and ScaleVLN's HM3D without depth-image pre-training, achieves an absolute gain of +8% in SR and SPL, demonstrating the strong transferability of our pre-trained navigator.

**REVERIE and SOON.** For high-level navigation tasks, as shown in Table 8, our method achieves notable improvements on the val unseen split for REVERIE and SOON, with +3.5% SPL over

Table 9: Performance of different instruction generators on R2R. † means reproduced results. Best results are marked in **bold blue**, second best in **bold**, and third best in underlined.

| Methods | R2R Validation Unseen | | | | | | |
|---|---|---|---|---|---|---|---|
| | SPICE↑ | SPICE-D↑ | Bleu-1↑ | Bleu-4↑ | CIDEr↑ | Meteor↑ | Rouge↑ |
| BT-speaker (Fried et al., 2018) | 18.9 | 25.1 | 68.2 | 26.3 | 37.9 | 21.7 | 48.0 |
| LandmarkSelect (Agarwal et al., 2019) | 19.7 | - | 54.8 | 15.9 | 13.2 | 23.1 | 35.7 |
| EnvDrop (Tan et al., 2019) | 21.8 | 28.0 | 72.3 | 27.1 | 41.7 | 23.6 | 49.0 |
| CCC (Wang et al., 2022a) | 21.4 | 27.8 | 70.8 | 27.2 | 46.1 | 23.1 | 47.7 |
| FOAM† (Dou & Peng, 2022) | 21.7 | 28.1 | 72.5 | 27.3 | 42.4 | 23.4 | 49.2 |
| KEFA (Zeng et al., 2023) | 23.4 | 29.3 | 73.8 | 28.3 | 42.7 | 24.4 | 50.3 |
| LANA (Wang et al., 2023a) | 22.6 | - | 73.6 | 28.9 | 45.7 | 23.7 | 49.8 |
| LANA+ (Wang et al., 2023b) | 22.8 | - | 73.2 | 29.5 | 46.0 | 24.1 | 49.6 |
| C-Instructor (Kong et al., 2024) | 21.2 | - | 71.3 | 26.6 | 44.7 | 23.9 | 47.3 |
| BEVInsructor (Fan et al., 2024) | 20.8 | - | 69.9 | 26.4 | 44.9 | 23.0 | 46.7 |
| SRDF (Ours, round 2) | 25.2 | 29.9 | 73.7 | **31.0** | **50.7** | 24.2 | **51.3** |
| SRDF (Ours, round 3) | **25.7** | **30.4** | **74.5** | 30.8 | **49.7** | **24.5** | **51.3** |
| SRDF (Ours, round 3 fine-tuned w/ $FD_4^G$) | **26.2** | **30.9** | **75.3** | **31.1** | 49.2 | **25.0** | **51.4** |

ScaleVLN and +10.0% SPL over AutoVLN (Chen et al., 2022c), respectively. These gains are especially impressive given that both AutoVLN and ScaleVLN (Wang et al., 2023d) used in-domain pre-training data, while ours comes from the out-of-domain R2R-style dataset. This demonstrates that our high-quality data not only improves fine-grained instruction-following but also enhances goal-finding, likely due to diverse stopping guidance. Surprisingly, our model achieves results comparable to the pre-exploration agent RREX-Bot (Sigurdsson et al., 2023) on REVERIE (-0.6% SR), and even surpasses it on SOON (+1.4% SR), showing our model's robust goal-finding ability.

**CVDN.** Our method also improves the previous best on the val unseen set of CVDN by a large margin (+1.51 meters) in Table 8, showing that our model can be generalized to different instruction styles, likely due to learning strong landmark alignment ability, which can be shared across tasks.

**RxR-English and R4R.** For long-horizon (and fine-grained) VLN tasks including RxR-en and R4R, our navigator also shows strong results on the val unseen split, surpassing previous SoTAs by a large margin shown in Table 8. It's worth noting that our results on R4R provide a very strong improvement of +16.6% SR. This shows our highly aligned data facilitates learning step-by-step instruction following even with very long trajectories.

**R2R Instruction Generation.** We also compare our instruction generator with previous SoTAs for path-to-instruction generation task on R2R val unseen split in Table 9. Thanks to our navigator-filtered high-quality data, our round 2 generator has already beat the previous SoTA significantly, with + 1.8 SPICE and + 4.6 CIDEr. The stronger data generated in the third round results in an additional +0.5 SPICE improvement while keeping other scores comparable or better. This substantial strong performance can even be further enhanced by incorporating $FD_4^G$ for fine-tuning, leading to a +0.5 improvement in SPICE and better results across five additional metrics, indicating the importance of high-quality data in instruction generator training.

## 5 CONCLUSION

In this work, we introduce a fully automatic self-refining data flywheel to construct a substantially high-quality VLN dataset for augmentation. We propose to iteratively refine the data with the navigator and instruction generator working in tandem—the navigator filtering high-quality data to train a better instruction generator and the instruction generator regenerating better instructions to train a better navigator, ultimately producing both a strong navigator, instruction generator and a high-quality VLN dataset. We thoroughly analyzed the impact of each component in the flywheel, demonstrating that our approach significantly surpasses state-of-the-art methods across multiple VLN benchmarks, covering various instruction styles (R2R, CVDN, REVERIE, SOON), trajectory lengths (R4R, RxR-English), and control spaces (R2R-CE), as well as instruction generation task on R2R. Our self-refining flywheel provides a novel, scalable solution to the data bottleneck in VLN, highlighting the crucial role of instruction quality and alignment in training embodied agents. This method has the potential to drive future advancements in embodied navigation models and paves the way for exploring more sophisticated tasks that rely on high-quality, dynamic, and scalable data.

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

# A    APPENDIX

We first present additional implementation details of our experiments in Section A.1, including specifics of the generator, navigator model, and training procedures. Section A.2 illustrates the SRDF pipeline with pseudo code, while Section A.4 provides further details and experiments regarding our trajectory-encoding template design. Details of datasets and evaluation metrics are provided in Section A.5, with more comprehensive downstream results shown in Section A.6. Finally, Section A.7 presents detailed examples of our generated instructions in comparison with other baselines.

## A.1    ADDITIONAL IMPLEMENTATION DETAILS

**Navigator.**    We use DUET model (Chen et al., 2022b) as our navigator, which integrates global and local information through a dual-scale graph transformer architecture. This architecture processes both high-level scene representations as well as detailed local features simultaneously, enhancing the model's capability to interpret language instructions in complex visual contexts. By constructing a topological map in the meanwhile, DUET extends the navigational action space from the current viewpoint to all navigable directions encountered, thus improving planning and error correction. The model employs attention mechanisms to balance global scene contexts with local observations, significantly improving navigation accuracy towards targets based on natural language commands.

**Generator.**    We use Mantis (Jiang et al., 2024) as our base model for instruction generation. Mantis comprises a SigLIP vision encoder (Zhai et al., 2023), a multimodal projector, and a LLaMA-3-8B-Instruct language model backbone (Dubey et al., 2024). It is first pre-trained on multimodal projector data using LLaVa pre-training, followed by fine-tuning on the multi-image interleaved Mantis-Instruct dataset.For our task, we initialize the instruction-tuned model, Mantis-8B-siglip-llama3, and apply updates only to the added LoRA layers in the language backbone.

**Training Details.**    The navigator is trained using similar objectives as in  (Wang et al., 2023d), including Masked Language Modeling and Single Action Prediction. We initialize our model with LXMERT (Tan & Bansal, 2019) for the first and second rounds, and with our round-2 pre-trained model for the third round. During generator training, all parameters except the injected LoRA layers are frozen, and only the LoRA layers are fine-tuned. For navigator training, we initialize the instruction generator with Mantis in the first and second rounds and use the round-2 trained generator directly for round 3. Additionally, for downstream task supervisions, we encourage the model to go back to the viewpoint on the GT path yielding best nDTW to the current progress for R4R and RxR-English as their trajectories are not shortest-path, while we use shortest-path supervision as teacher action for other tasks.

## A.2    PSEUDO CODE OF SRDF

Alg. 1 provides detailed pseudo code for our Self-Refining Data Flywheel (SRDF), illustrating the process described in Section 3.2. Specifically, the pipeline starts with training an initial instruction generator using seed data. The generator creates training data for the navigator, which is then used to train a base navigator. The navigator, in turn, filters high-quality data to further improve both itself and the generator in subsequent rounds. This iterative process continues for $T$ iterations, refining data quality and improving model robustness with each cycle.

## A.3    SFT DATA TEMPLATE

We use the template shown in Figure 3 to construct our SFT data for fine-tuning the instruction generator, which encodes a trajectory into an interleave image-action sequence. At each viewpoint, we include key views, such as the view when arriving at the viewpoint and the view when leaving it. For each view, we also append the corresponding action in raw text after the image tokens, creating a multi-image interleaved format to effectively encode the trajectories. The output is the corresponding instruction of the input trajectory.

---

**[Model Input]**
This is a navigation trajectory consists of several image sequences:
(Viewpoint 1: Image:`<image>`, Action: left (135.71 degree) and up (0.00 degree)),
(Viewpoint 1: Image:`<image>`, Action: forward),
(Viewpoint 2: Image:`<image>`, Action: left (33.14 degree) and up (0.00 degree)),
...
(Viewpoint 5: Image:`<image>`, Action: forward),
(Viewpoint 6: Image:`<image>`, Action: stop).
Could you give me its corresponding navigation instruction in details?

---

**[Model Output]**
Start by exiting the bedroom into the hallway. Turn right and proceed down the hall until you reach the first door on your left. Enter this room, which is the library. Once inside, walk towards the large bookshelf directly across from the doorway. Stop there and face towards the window on the right side of the room.

---

Figure 3: Interleaved trajectory prompt and expected response for training our instruction generator. The LLM's response is the corresponding instruction of this path.

## A.4 ADDTIONAL EXPERIMENTS

**Fourth-Round Generator-Training.** Our method demonstrates the potential for continued performance improvement with additional refinement rounds. To illustrate this, we conducted a fourth-round generator training experiment (which is equivalent to the "round 3 w/ $FD_4^G$" results in Table 9) Following a similar procedure, we generated $FD_4^G$, trained the generator using this data, and observed consistent performance gains over the third round. Results in Table 10 show a new state-of-the-art (SoTA) SPICE score of 26.2, marking an improvement of +2.8 over the previous SoTA (as shown in Table 9). These results highlight the scalability and effectiveness of our approach, indicating that with sufficient computational resources and time, further rounds of improvement can be sustained.

**Effect of Different Scoring Metrics in Generator Training.** We conducted additional experiments to demonstrate that classical methods for language model self-improvement face limitations in the VLN context without reliable feedback from the navigator. Specifically, we evaluated two approaches: (1) self-score, a self-rewarding method where the language model scores its own outputs, and (2) CLIP-score, which uses an external tool (CLIP) to provide similarity scores. In these exper-

---

**Algorithm 1** Pipeline of Self-Refining Data Flywheel (SRDF)

---

**Require:** Seed data $D_{Seed}$ (Human-annotated), Unlabelled trajectories $D_{Traj}$, Total iterations $T$.
1: Train base instruction generator $G_1$ with $D_{Seed}$.
2: Use $G_1$ to generate nav-training data $D_1^N$ and gen-training data $D_2^G$ for $D_{Traj}$.
3: /* $D_1^N$ is generated via random sampling while $D_1^G$ via greedy decoding
4: Train base navigator $N_1$ with $D_1^N$.
5: Use $N_1$ to filter high-quality subsets $FD_2^G$ from $D_2^G$ and $FD_{<2}^N$ from $D_1^N$.
6: **for** each iteration $t$ $(1 < t \leqslant T)$ **do**
7:     /* Note: Seed data $D_{Seed}$ is used in training stages of both $G_t$ and $N_t$ but omitted for simplicity
8:     Train generator $G_t$ with $FD_t^G$.
9:     Use $G_t$ to generate nav-training data $ND_t^N$ for $LD_t^N$ and gen-training data $D_{t+1}^G$ for $D_{Traj}$.
10:     /* $ND_t^N$ is generated via random sampling while $D_{t+1}^G$ via greedy decoding
11:     Combine $ND_t^N$ and $FD_{<t}^N$ to form $D_t^N$.
12:     Train navigator $N_t$ with $D_t^N$.
13:     Use $N_t$ to filter high-quality subsets $FD_{t+1}^G$ from $D_{t+1}^G$ and $FND_t^N$ from $ND_t^N$.
14:     Combine $FND_t^N$ and $FD_{<t}^N$ to form $FD_{<t+1}^N$.
15: **end for**

---

iments, conducted during round 1, instructions were scored and the top 300K instructions filtered using either (1) or (2) were used to train the instruction generator in round 2. The results in Table 11 showed that neither self-scores nor CLIP scores yielded significant improvement over the round 1 baseline. In contrast, our navigator-filtering method using nDTW demonstrated substantial gains, highlighting the challenges of providing effective feedback and emphasizing the effectiveness of our approach.

Table 10: Generator results in the additional fourth round.

| Method | SPICE↑ | Bleu-1↑ | Bleu-4↑ | CIDEr↑ | Meteor↑ | Rouge↑ |
|---|---|---|---|---|---|---|
| Baseline | 21.8 | 72.5 | 27.7 | 42.2 | 23.6 | 49.0 |
| Ours (round 1) | 23.7 | 71.4 | 29.5 | 46.5 | 23.1 | 50.2 |
| Ours (round 2) | 25.2 | 73.7 | 31.0 | **50.7** | 24.2 | 51.3 |
| Ours (round 3) | 25.7 | 74.5 | 30.8 | 49.7 | 24.5 | 51.3 |
| Ours (round 4) | **26.2** | **75.3** | **31.1** | 49.2 | **25.0** | **51.4** |

Table 11: Second-round generator performance with different scorers.

| round | scorer | SPICE↑ | Bleu-1↑ | Bleu-4↑ | CIDEr↑ | Meteor↑ | Rouge↑ |
|---|---|---|---|---|---|---|---|
| round 1 | - | 23.7 | 71.4 | 29.5 | 46.5 | 23.1 | 50.2 |
| round 2 | Self-score | 23.6 | 71.3 | 29.4 | 46.4 | 23.5 | 50.3 |
| round 2 | CLIP-score | 23.9 | 70.6 | 30.0 | 48.6 | 23.1 | 50.4 |
| round 2 | navigator-nDTW | **25.2** | **73.7** | **31.0** | **50.7** | **24.2** | **51.3** |

Table 12: Two-round generator performance with mPLUG-Owl.

| Method | SPICE↑ | Bleu-1↑ | Bleu-4↑ | CIDEr↑ | Meteor↑ | Rouge↑ |
|---|---|---|---|---|---|---|
| Baseline | 21.8 | 72.5 | 27.7 | 42.2 | 23.6 | 49.0 |
| mPLUG-owl (round 1) | 22.7 | 70.3 | 28.0 | 44.4 | 23.0 | 49.1 |
| mPLUG-owl (round 2) | **24.3** | **72.2** | **29.1** | **45.2** | **23.7** | **50.0** |

**Effect of Different MLLM** To demonstrate that our model-boosting process is not reliant on Mantis (Jiang et al., 2024), we conducted additional experiments with a weaker multimodal large language model (MLLM), mPLUG-Owl-7B (Ye et al., 2023). Using the same methodology, we applied the flywheel process and completed the first two rounds of generator training. In Table 12, we observed a significant improvement in the round 2 generator's performance when trained on its data filtered by the navigator, highlighting the navigator's critical role in enhancing the generator. Given that the reciprocal improvement of the navigator by the generator has been validated in prior work using the Speaker-Follower framework, these results strongly support our assertion that the model-boosting process is robust and not tied to a specific MLLM.

**Effect of Different Encoding Formats.** In our previous discussion, we hypothesized that the interleaved image-text understanding ability is crucial for training instructor generator based on pre-trained MLLMs. Some prior works (Li & Bansal, 2023; Kong et al., 2024) use only image information to build an image sequence for fine-tuning the VLM, but we argue that this approach loses important directional clues. Additionally, if action information is added without an interleaved format (i.e., an action sequence followed by an image sequence), the model may struggle to reason effectively between the two sequences.

We verify this hypothesis in Table 13, using two baselines: (1) an image-only sequence (Figure 3 without action descriptions) and (2) an action sequence followed by an image sequence (Figure 3 with actions listed after the image sequence). Our results show that using only an image sequence leads to much lower performance, primarily because the model finds it difficult to infer actions between key frames. For the image-sequence + action-sequence format, it still underperforms compared to our interleaved image-text sequence template, likely due to challenges in reasoning across

Table 13: Effect of different trajectory-encoding templates.

| Templates | R2R Validation Unseen | | | | | | |
|---|---|---|---|---|---|---|---|
| | SPICE↑ | SPICE-D↑ | Bleu-1↑ | Bleu-4↑ | CIDEr↑ | Meteor↑ | Rouge↑ |
| Image Seq. | 21.8 | 24.4 | 68.7 | 25.7 | 46.5 | 21.9 | 48.3 |
| Image Seq. + Action Seq. | 22.9 | 27.1 | 70.4 | 29.1 | 46.5 | 22.9 | 50.2 |
| Interleave Image-Action Seq. | **23.7** | **28.4** | **71.4** | **29.5** | 46.5 | **23.1** | **50.2** |

separate sequences. In contrast, the Interleaved Image-Action Sequence performs best, demonstrating its effectiveness in trajectory encoding, which is used in our experiments.

## A.5 DETAILS OF DATASETS AND EVALUATION METRICS

**Datasets.** We conduct our downstream experiments on 7 datasets listed below.

- **R2R**: Consists of 22k human-annotated navigational instructions, each describing a trajectory that traverses multiple rooms in MP3D. On average, an instruction contains 32 words, and each ground-truth path is formed by seven nodes with a total length of 10 meters.
- **REVERIE**: Inherits the trajectories in R2R but provides high-level instructions that describe a target object. The task for an agent is first to find the object and then localize it in the observation.
- **SOON**: Provides instructions describing target rooms and objects. The average length of instructions is 47 words. SOON does not provide object bounding boxes and requires the agent to predict object center locations in the panorama. We use an automatic object detector to obtain candidate object boxes. The length of expert paths ranges from 2 to 21 steps, with an average of 9.5 steps.
- **CVDN**: Provides dialogues between a navigator who tries to find a target by asking for guidance and an oracle with a privileged view of the best next step. The agent must find the way by interpreting the dialogue history.
- **R2R-CE**: Transfers the discrete trajectories in R2R to continuous 3D scans rendered by the Habitat simulator, where an agent can freely travel in the open space and interact with obstacles. The dataset contains 16k instruction-trajectory pairs after removing non-transferable paths.
- **RxR**: An extension of R2R that addresses shortest path biases and includes more object references. We use the English segment of RxR, which consists of 42,002 instructions, averaging 108 words per instruction.
- **R4R**: Created by concatenating adjacent paths in the Room-to-Room dataset. The ground-truth path is not the shortest path, encouraging the agent to follow the instructions to reach the target rather than exploit environment bias to navigate the shortest route.

**Detailed Evaluation Metrics.** (1) Success Rate (SR), which measures whether the agent stops within 3 meters of the target; (2) Success Rate Weighted by Path Length (SPL), which penalizes inefficient, longer paths; (3) Goal Progress (GP), which calculates the agent's progress toward the target; (4) Navigation Error (NE), which is the average distance between the agent's final position and the target in meters; (5) normalized Dynamic Time Warping (nDTW), which measures stepwise alignment between the ground truth and the agent-predicted path; (6) Success Rate Weighted by Dynamic Time Warping (sDTW); (7) Coverage weighted by Length Score (CLS); (8) Remote Grounding Success (RGS), the proportion of successfully executed instructions; and (9) RGS Penalized by Path Length (RGSPL). Metrics (7) to (9) are used only in the detailed results provided in the appendix, with (8) and (9) specifically evaluating object grounding in REVERIE and SOON.

## A.6 DETAILED RESULTS

In this section, we present detailed results for our downstream navigators, across multiple datasets: R2R (Table 18), R4R(Table 14), RxR-English (Table 15), REVERIE (Table 17), and SOON datasets (Table 16). Specifically, on the R2R dataset, our model not only demonstrates improved generalizability to unseen environments but also achieves higher success rates and SPL in seen environments, surpassing previous state-of-the-art (SoTA) approaches by over 3% in SR and 4% in SPL. Our method achieved more than a 10% improvement in both SR and sDTW on the R4R dataset. Besides, on the RxR English dataset, our approach significantly enhances SPL and sDTW in addition to SR and nDTW, elevating the state-of-the-art SPL to 69.2% and sDTW to 66.3%. Lastly, on REVERIE

and SOON datasets, our navigator not only enhances the agents' navigation performance significantly but also substantially improves their grounding capabilities, improving RGSPL by 0.4% on REVERIE test set, and 1.7% on SOON test set compared with previous SoTA approaches. Notably, our navigator relies solely on pretraining with Masked Language Modeling (MLM), and Single Action Prediction (SAP) objectives on R2R datasets and the augmentation dataset collected with our data flywheel. This is in contrast to other approaches that additionally employ an Object Grounding (OG) objective. The superior performance indicates the strong instruction-trajectory alignment in our high-quality data, which is crucial for effectively learning object grounding from scratch during fine-tuning.

Table 14: Comparison of single-run performance on R4R dataset.

| Methods | Validation Unseen | | | | |
|---------|------|------|------|--------|--------|
|         | NE↓  | SR↑  | CLS↑ | nDTW↑  | sDTW↑  |
| HAMT (Chen et al., 2021) | 6.09 | 44.6 | 57.7 | 50.3 | 31.8 |
| LOVIS (Zhang & Kordjamshidi, 2022b) | 6.07 | 45.0 | 45.0 | 43.0 | 23.0 |
| VLN-Trans (Zhang & Kordjamshidi, 2023) | **5.87** | 46.0 | 45.0 | - | 25.0 |
| PanoGen (Li & Bansal, 2023) | 6.02 | **47.8** | - | - | - |
| BSG (Liu et al., 2023) | 6.12 | 47.0 | 59.0 | 53.0 | **34.0** |
| NavHint (Zhang et al., 2024a) | 6.04 | 46.0 | 45.0 | - | 25.0 |
| VER (Liu et al., 2024) | 6.10 | 47.0 | **61.0** | **54.0** | 33.0 |
| SRDF (Ours) | **4.21** | **64.4** | **61.0** | **56.5** | **44.6** |

Table 15: Comparison of single-run performance on RxR English dataset. † indicates the results are reproduced using their officially released checkpoints.

| Methods | Validation Seen | | | | | Validation Unseen | | | | |
|---------|------|------|-------|--------|--------|------|------|-------|--------|--------|
|         | NE↓  | SR↑  | SPL↑  | sDTW↑  | nDTW↑  | NE↓  | SR↑  | SPL↑  | sDTW↑  | nDTW↑  |
| Landmark-RxR (He et al., 2021) | - | 64.1 | - | 53.1 | 63.4 | - | 40.1 | - | 27.5 | 40.3 |
| HAMT† (Chen et al., 2021) | - | 59.4 | - | 50.9 | 65.3 | - | 56.5 | - | 48.3 | 63.1 |
| MARVAL (Kamath et al., 2022) | 3.31 | 74.0 | - | 66.7 | **77.5** | 4.47 | 64.7 | - | 57.1 | 70.5 |
| PRET† (Lu et al., 2024) | **2.68** | 77.1 | 71.8 | 67.1 | **77.5** | **3.36** | 71.0 | 63.6 | **63.5** | **70.9** |
| MAGIC-L (Wang et al., 2024b) | - | **81.3** | **77.5** | **69.2** | 76.6 | - | **72.9** | **65.4** | 58.7 | 68.1 |
| BEVBert† (An et al., 2023a) | - | - | - | - | - | 4.2 | 66.7 | 61.1 | 57.0 | 68.6 |
| GOAT (Wang et al., 2024a) | - | 74.1 | 68.1 | 61.4 | 71.0 | - | 68.2 | 61.7 | 56.6 | 67.1 |
| SRDF (Ours) | **1.95** | **82.9** | **77.7** | **75.6** | **83.4** | **2.61** | **78.8** | **69.2** | **66.3** | **74.4** |

Table 16: Comparison of single-run performance on SOON dataset.

| Methods | Validation Unseen | | | | Test Unseen | | | |
|---------|------|------|------|-----------|------|------|------|-----------|
|         | Navigation | | | Grounding | Navigation | | | Grounding |
|         | OSR↑ | SR↑  | SPL↑ | RGSPL↑    | OSR↑ | SR↑  | SPL↑ | RGSPL↑    |
| DUET (Chen et al., 2022b) | 50.9 | 36.3 | 22.6 | 3.8 | 43.0 | 33.4 | 21.4 | 4.2 |
| AutoVLN (Chen et al., 2022c) | 53.2 | **41.0** | **30.7** | 4.1 | 48.7 | 40.4 | **27.8** | 5.1 |
| KERM (Li et al., 2023b) | 51.6 | 38.1 | 23.2 | 4.0 | - | - | - | - |
| NaviLLM (Zheng et al., 2024) | - | 38.3 | 29.2 | - | - | 35.0 | 26.3 | - |
| GOAT (Wang et al., 2024a) | **54.7** | 40.4 | 28.1 | **5.9** | 50.6 | 40.5 | 25.2 | **6.1** |
| SRDF (Ours) | **59.6** | **50.3** | **41.7** | **5.1** | **51.6** | **46.6** | **37.9** | **8.4** |

Table 17: Comparison of single-run performance on REVERIE datasets.

| Methods | Validation Unseen | | | | | Test Unseen | | | | |
|---|---|---|---|---|---|---|---|---|---|---|
| | Navigation | | | Grounding | | Navigation | | | Grounding | |
| | OSR↑ | SR↑ | SPL↑ | RGS↑ | RGSPL↑ | OSR↑ | SR↑ | SPL↑ | RGS↑ | RGSPL↑ |
| HAMT (Chen et al., 2021) | 36.8 | 33.0 | 30.2 | 18.9 | 17.3 | 33.4 | 30.4 | 26.7 | 14.9 | 13.1 |
| DUET (Chen et al., 2022b) | 51.1 | 47.0 | 33.7 | 32.2 | 23.0 | 56.9 | 52.5 | 36.1 | 31.9 | 22.1 |
| BEVBert (An et al., 2023a) | 56.4 | 51.8 | 36.4 | 34.7 | 24.4 | 57.3 | 52.8 | 36.4 | 32.1 | 22.1 |
| AutoVLN (Chen et al., 2022c) | 62.1 | 55.9 | 40.9 | 36.6 | 26.8 | - | - | - | - | - |
| KERM (Li et al., 2023b) | 55.2 | 50.4 | 35.4 | 34.5 | 24.5 | 57.6 | 52.4 | 39.2 | 32.4 | 23.6 |
| BSG (Liu et al., 2023) | 58.1 | 52.1 | 35.6 | 35.4 | 24.2 | 62.8 | 56.5 | 38.7 | 33.2 | 22.3 |
| ScaleVLN (Wang et al., 2023d) | **63.9** | 57.0 | 41.8 | - | - | 62.7 | 56.1 | 39.5 | - | - |
| MiC (Qiao et al., 2023b) | 62.4 | 57.0 | 43.6 | 37.5 | **28.7** | 62.4 | 55.7 | 42.0 | 35.3 | 26.2 |
| NaviLLM (Zheng et al., 2024) | 53.7 | 44.6 | 36.6 | - | - | 56.2 | 43.5 | 34.4 | - | - |
| VER (Liu et al., 2024) | 61.1 | 56.0 | 39.7 | 33.7 | 23.7 | 62.2 | 56.8 | 38.8 | 33.9 | 23.2 |
| GOAT (Wang et al., 2024a) | - | 53.4 | 36.7 | **38.4** | 26.1 | - | 57.7 | 40.5 | **38.3** | **26.7** |
| VLN-Colipot (Qiao et al., 2024) | 62.6 | **57.4** | **43.6** | **38.9** | **29.8** | 63.3 | 57.8 | 42.3 | 36.6 | 26.6 |
| SRDF (Ours) | **72.2** | **60.4** | **45.4** | 37.3 | 27.8 | **66.2** | **61.4** | **47.7** | 35.6 | **27.1** |

Table 18: Comparison of single-run performance on R2R dataset.

| Methods | Validation Seen | | | | Validation Unseen | | | | Test Unseen | | | |
|---|---|---|---|---|---|---|---|---|---|---|---|---|
| | NE↓ | OSR↑ | SR↑ | SPL↑ | NE↓ | OSR↑ | SR↑ | SPL↑ | NE↓ | OSR↑ | SR↑ | SPL↑ |
| Human | - | - | - | - | - | - | - | - | 1.61 | 90 | 86 | 76 |
| Seq2Seq (Anderson et al., 2018b) | 6.01 | 53 | 39 | - | 7.81 | 28 | 21 | - | 7.85 | 27 | 20 | - |
| Speaker Follower (Fried et al., 2018) | 3.36 | 74 | 66 | - | 6.62 | 45 | 36 | - | 6.62 | - | 35 | 28 |
| RCM (Wang et al., 2019) | 3.53 | 75 | 67 | - | 6.09 | 50 | 43 | - | 6.12 | 50 | 43 | 38 |
| EnvDrop (Tan et al., 2019) | 3.99 | - | 62 | 59 | 5.22 | - | 52 | 48 | 5.23 | 59 | 51 | 47 |
| PREVALENT (Hao et al., 2020) | 3.67 | - | 69 | 65 | 4.71 | - | 58 | 53 | 5.30 | 61 | 54 | 51 |
| EntityGraph (Hong et al., 2020) | 3.47 | - | 67 | 65 | 4.73 | - | 57 | 53 | 4.75 | 61 | 55 | 52 |
| NvEM (An et al., 2021) | 3.44 | - | 69 | 65 | 4.27 | - | 60 | 55 | 4.37 | 66 | 58 | 54 |
| SSM (Wang et al., 2021) | 3.10 | 80 | 71 | 62 | 4.32 | 73 | 62 | 45 | 4.57 | 70 | 61 | 46 |
| AirBert (Guhur et al., 2021) | 2.68 | - | 75 | 70 | 4.10 | - | 62 | 56 | 4.13 | - | 62 | 57 |
| VLN↻BERT (Hong et al., 2021) | 2.90 | - | 72 | 68 | 3.93 | - | 63 | 57 | 4.09 | 70 | 63 | 57 |
| HAMT (Chen et al., 2021) | 2.51 | - | 76 | 72 | 2.29 | - | 66 | 61 | 3.93 | 72 | 65 | 60 |
| EnvMix (Liu et al., 2021a) | 2.48 | - | 75 | 72 | 3.89 | - | 64 | 58 | 3.87 | 72 | 65 | 59 |
| SnapEnsemble (Qin et al., 2021) | - | - | - | - | 3.63 | - | 67 | 60 | 3.82 | - | 65 | 60 |
| EXOR (Zhang & Kordjamshidi, 2022a) | - | - | 60 | 58 | - | - | 52 | 49 | - | - | 49 | 46 |
| SEvol (Chen et al., 2022a) | 3.56 | - | 67 | 63 | 3.99 | - | 62 | 57 | 4.13 | - | 62 | 57 |
| MARVAL (Kamath et al., 2022) | 2.99 | - | 73 | 69 | 4.06 | - | 65 | 61 | 4.18 | 67 | 62 | 58 |
| LOVIS (Zhang & Kordjamshidi, 2022b) | 2.40 | - | 77 | 72 | 3.71 | - | 65 | 59 | 4.07 | - | 63 | 58 |
| HOP+ (Qiao et al., 2023a) | 2.33 | - | 78 | 73 | 3.49 | - | 67 | 61 | 3.71 | - | 66 | 60 |
| TD-STP (Zhao et al., 2022) | 2.34 | 83 | 77 | 73 | 3.22 | 76 | 70 | 63 | 3.73 | 72 | 67 | 61 |
| DUET (Chen et al., 2022b) | 2.28 | 86 | 79 | 73 | 3.31 | 81 | 72 | 60 | 3.65 | 76 | 69 | 59 |
| ScaleVLN (Wang et al., 2023d) | 2.12 | 87 | 81 | 75 | **2.09** | 88 | 81 | 70 | **2.27** | 86 | 80 | 70 |
| BEVBert (An et al., 2023a) | 2.17 | 88 | 81 | 74 | 2.81 | 84 | 75 | 64 | 3.13 | 81 | 73 | 62 |
| VLN-Trans (Zhang & Kordjamshidi, 2023) | 2.45 | - | 77 | 72 | 3.34 | - | 69 | 63 | 3.94 | - | 66 | 60 |
| Lily (Lin et al., 2023) | - | - | - | - | 2.90 | - | 74 | 62 | 3.44 | - | 72 | 60 |
| NaviLLM (Zheng et al., 2024) | - | - | - | - | 3.51 | - | 67 | 59 | 3.71 | - | 68 | 60 |
| NavHint (Zhang et al., 2024a) | - | - | - | - | 3.23 | - | 69 | 65 | 4.00 | - | 65 | 60 |
| NavGPT-2 (Zhou et al., 2024a) | 2.84 | 83 | 74 | 63 | 2.84 | 84 | 74 | 61 | 3.33 | 80 | 72 | 60 |
| SAME (Zhou et al., 2024b) | - | - | - | - | 2.73 | - | 76 | 66 | 3.03 | - | 74 | 64 |
| VER (Liu et al., 2024) | - | - | - | - | 2.80 | - | 76 | 65 | 2.74 | - | 76 | 66 |
| MAGIC-L (Wang et al., 2024b) | **1.73** | 89 | 84 | 79 | 2.22 | 86 | 79 | **70** | 2.75 | 82 | 77 | 69 |
| GOAT (Wang et al., 2024a) | 1.79 | **89** | 84 | 79 | 2.40 | 85 | 78 | 68 | 3.04 | 80 | 75 | 65 |
| SRDF (Ours) | **1.54** | **91** | **87** | **83** | **1.62** | **90** | **86** | **79** | **1.82** | **89** | **85** | **78** |

## A.7 QUALITATIVE CASE STUDY OF GENERATED INSTRUCTIONS

In Figure 4 and 5, we visualized some examples of our generated instructions, and compare them with Prevalent (Hao et al., 2020) and ScaleVLN (Wang et al., 2023d) baselines. All the example trajectories in Figure 4, and Figure 5 (a), (b) are collected using recovered environment images from ScaleVLN (Wang et al., 2023d), while Figure 5 (c), (d) are from MP3D environments.

**Rare-Room/Landmark Recognition Ability.** Figure 4 demonstrates the strong image-text understanding capability of our instruction generator. Specifically, our generator can recognize rare objects, such as *dentist chair/room* or *a grandfather clock*, thanks to our interleaved image-action trajectory-encoding design. This design preserves the original abilities of the pretrained MLLM while effectively encoding trajectories to generate instructions with rich and accurate landmarks. In contrast, the baseline instruction generator fails to capture these rare concepts due to its from-scratch training paradigm. Instead, it only generates some general landmarks with weak clues.

**Detailed Object-Describing Ability.** Figures 4 (c) and (d) illustrate that our instruction generator can describe key objects along the path with greater detail. For instance, while the baseline mentions the *bar* and *the painting* successfully, our generator provides more specifics, such as *brown leather bar stool* and *large painting on the wall*. Such detailed descriptions are crucial for helping the navigator learn richer visual cues. Additionally, in Figure 4 (d), our generator performs slight spatial reasoning between objects, resulting in more precise stopping guidance – *the blue and white throw pillows on the right side of the couch*.

**Generalization to Outdoor Environments.** In Figure 5 (a), (b), we demonstrate the ability of our generator to produce some useful instructions for outdoor environments, even though the model is training using instructions from indoor environments. For instance, in (a), our generated instruction identifies *the glass doors leading outside*, which is more distinct the ScaleVLN's *table* – still a general landmark without a strong viewpoint-specific clue. In (b), the generator successfully identifies outdoor landmarks including *the car* and *the buches*, while the baseline only knows *walkway*.

**OCR Ability.** Surprisingly, our instruction generator demonstrates interesting OCR capabilities, as shown in Figures 5 (c) and (d). In example (c), the generator successfully identifies the words *cape* and *plug* on the wall, while in example (d), it even identifies a full sentence—*Let's start to redefine how work gets done*. This OCR ability is likely inherited from the pre-trained MLLM, and our fine-tuning approach effectively retains this capability, resulting in highly detailed and accurate guidance in the generated instructions.

**Idiomatic Expressions.** Our generator sometimes uses idiomatic expressions in its instructions. An example is shown in Figure 5 (c), Sample 2, where the generator says, *go past the desk then stop at the end of the rope*. The phrase *at the end of the rope* usually means that someone has reached the limit of their patience or endurance. In this context, however, it refers to reaching the farthest point that the navigator can proceed—likely the wall. This ability adds diversity to the instructing style, making the generated instructions more varied and engaging.

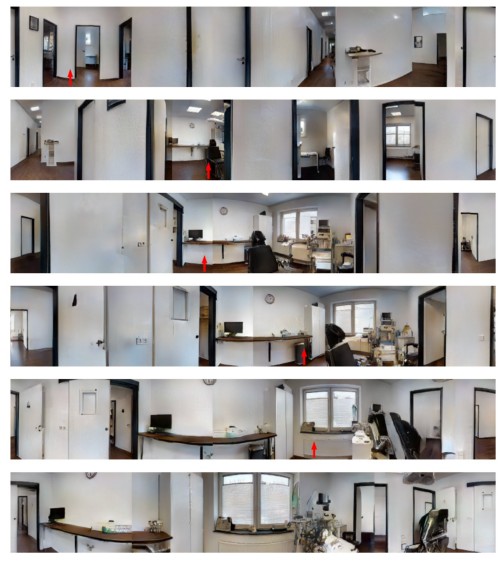

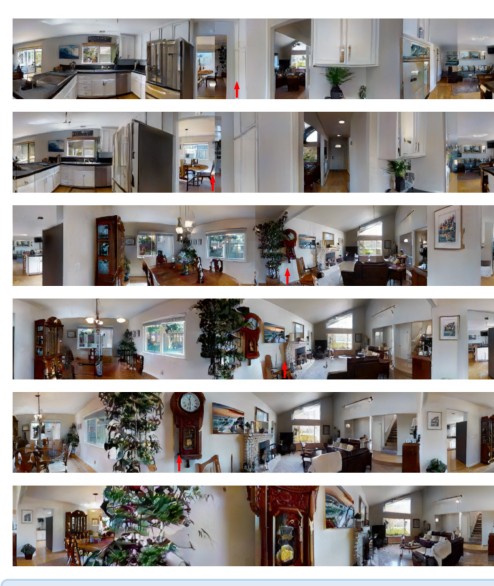

**ScaleVLN-EnvDrop**:
turn around and walk through the doorway. turn left and walk past the table and chairs. turn right and walk into the room. wait near the sink.

**Ours**:
walk through the door on the left and into the dentist room. walk past the dental chair. turn right and stop in front of window near the closet.

(a) Rare room identification

**ScaleVLN-EnvDrop**:
walk through the kitchen and into the dining room . walk past the dining table and stop in front of the stairs.

**Ours**:
walk through the hallway and into the dining room. walk past the table and chairs and into the living room. walk past the rocking chair and stop in front of the grandfather clock.

(b) Rare landmark identification

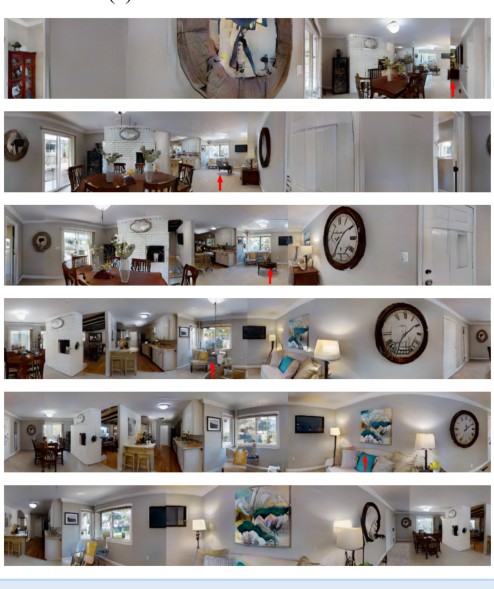

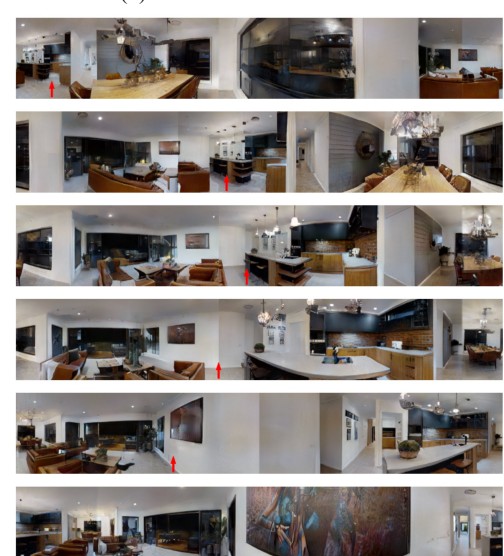

**ScaleVLN-EnvDrop**:
walk past the dining room table and chairs and turn right. walk past the dining room table and chairs and turn right. stop in front of the couch.

**Ours**:
walk down the hallway past the dining room table and chairs. continue into the living room and walk past the couch. stop in front of the blue and white throw pillows on the right side of the couch.

(c) Object relationships

**ScaleVLN-EnvDrop**:
walk past the dining room table and chairs and turn left. walk past the bar and turn left. stop in front of the painting.

**Ours**:
walk past the dining table and chairs. walk past the kitchen island. turn left and walk past the brown leather bar stools. turn left and stop in front of the large painting on the wall.

(d) Detailed descriptions

Figure 4: Visualization of generated instructions.

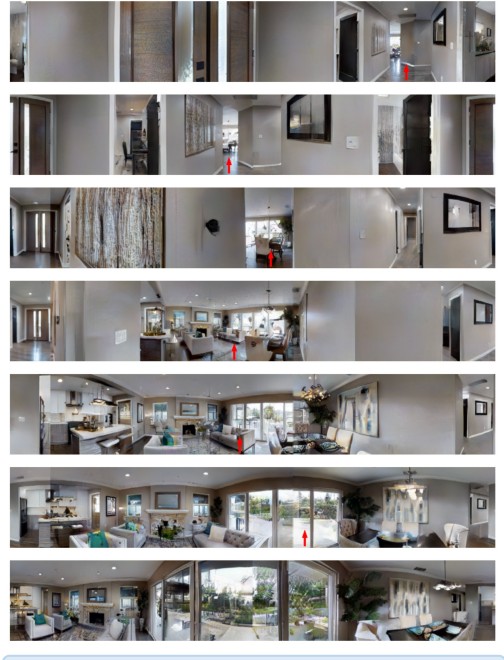

**ScaleVLN-EnvDrop**:
walk down the hallway and turn right. walk past the dining table and stop in front of the table.

**Ours**:
turn right and walk down the hallway. go through the door at the end of the hallway. walk past the dining room table and chairs. walk past the couch and stop in front of the glass doors leading outside.

(a) Ourdoor environment identification

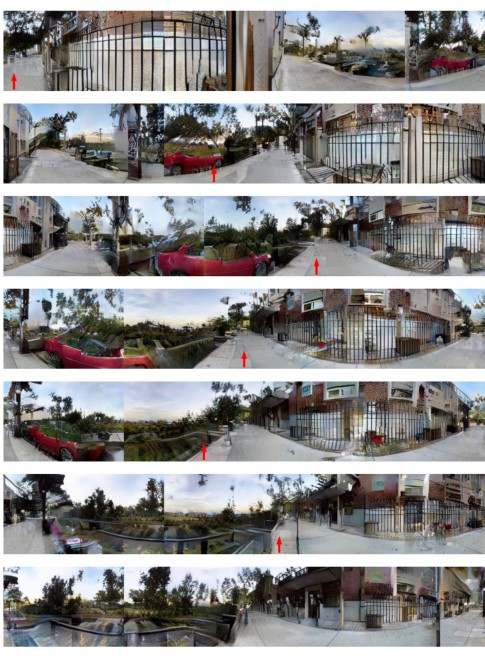

**ScaleVLN-EnvDrop**:
walk down the walkway and turn right . walk down the walkway and stop at the end of the walkway.

**Ours**:
walk past the car. turn left and walk down the sidewalk. wait by the bushes.

(b) Ourdoor environment generalization

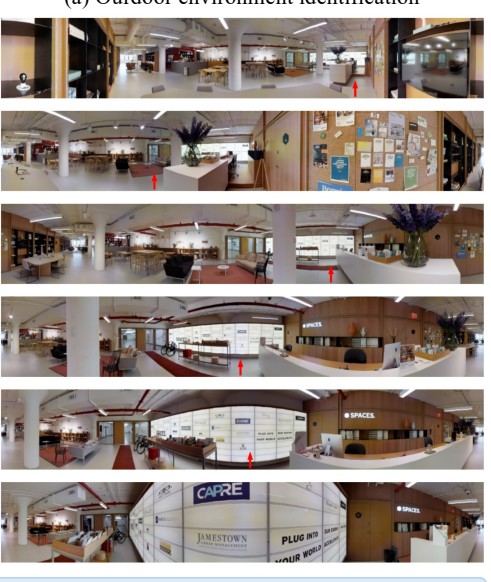

**Prevalent**:
turn around, go forward, take a right when you get to the table walk into the center room and stop.

**Ours (Sample 1)**:
go down the hallway and past the desk. turn right at the end of the hallway and stop in front of the wall with the word "cape" on it and "plug" on the wall.

**Ours (Sample 2)**:
go straight, then turn left and go around the pillar. then turn right and go straight. go passed the desk then stop at the end of the rope.

(c) OCR ability + Idiomatic expressions

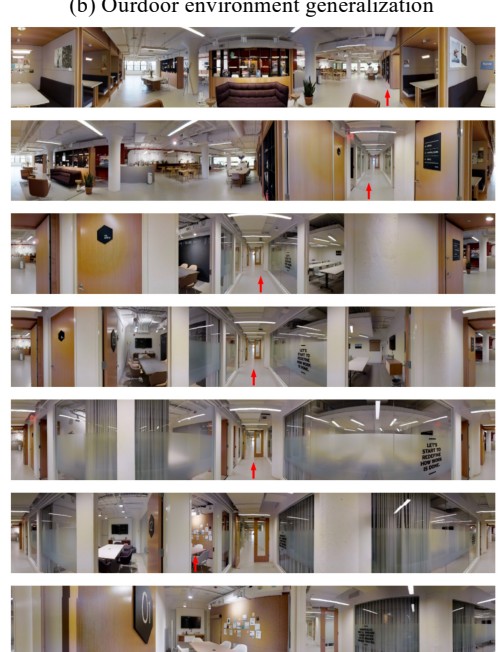

**Prevalent**:
leave the office and take a right. go down the hall and stop in the first doorway straight in a ballroom.

**Ours**:
walk around the table and chairs, and turn into a hallway to the right with a sign on the door that read "Let's start to redefined how work gets done". walk down the hallway until you see the door on the left that read "01".

(d) OCR ability

Figure 5: Visualization of generated instructions.

