# OpenReview forum: "Bootstrapping Language-Guided Navigation Learning with Self-Refining Data Flywheel"
_ICLR.cc/2025/Conference — ICLR 2025 Poster_

### Official Review · Reviewer_JHxr · 2024-10-26

**Soundness:** 3
**Presentation:** 3
**Contribution:** 3
**Rating:** 8
**Confidence:** 4

**Summary:**

This paper presents a self-refining data flywheel (SRDF) approach to generate the high quality the instruction data for vision-language navigation task. It proposes a generator and navigator iteratively tuned to train each other. First, a navigator and a generator are trained on the base corpus, then the navigator serves as a filter for the generator to collect high-quality synthetic data, then refine the navigator with synthetic data iteratively. Empirical experiments on several standard vision-language navigation datasets (R2R, CVDN, R4R, REVERIE etc.) show very promising results.

**Strengths:**

1. An interesting paper to show that iteratively tuning the generator and navigator works in the vision-language navigation, and also demonstrates the state-of-the-art results on several tasks. The empirical results are impressive.

**Weaknesses:**

1. It looks this work (proposing iteratively tuning generator and navigator), is a upgraded version of Speaker-Follower model, which you can think it is only the first iteration of the model. Could you discuss in more details about the relevance and difference with Speaker-Follower in the paper?

**Questions:**

Some questions -

1. For the table 5, the number of instructions, did you use multiple instructions encoders (with one path trajectory) as input? for example, N=3, do you treat (3 instructions, one trajectory) as one instance? or (3 instructions, one trajectory) as three (1 instructions, one trajectory) instance? This is quite different.  If your input is multiple instruction encoders (or shared encoder for multiple instructions as input), this is quite different with one instruction input.

Could you clarify the exact input and encoding process for this part?

2. And also for the main table 7, how many instructions per path trajectory in your setting?

3. Did you try this method on a different environment? rather than on R2R environments. Nearly all the tasks of this work are built on the R2R environments.  One question here is the generalization of the proposed method, if you could try out-of-domain environments, for example, ALFRED task?

---

> ### Author Response · Authors · 2024-11-21
> **Response to Reviewer 4 (JHxr)**
>
> We thank Reviewer 4 (JHxr) for acknowledging the contribution of our paper and for the thoughtful comments. Please see our response to the comments below.
>
> >**Relationship to speaker-follower models. (R4W1)**
>
> We want to point out that the speaker-follower model is a widely used framework in VLN for data augmentation, with many previous works contributing distinct innovations within this scope (see Table 9 for speaker-related work and Table 2 “Generated” for data generation using speakers). Our approach builds on this framework but introduces a novel iterative process for mutual improvement between the navigator and generator to self-improve the VLN data quality.
>
> Additionally, we also introduce significant advancements to VLN speaker research, starting with a strong intuition (Section 3.2) on text generalization, interleaved and multi-image context understanding ability to use what kind of MLLM, supported by well-designed input strategies (Figure 3, Table 10) to build a strong generator. Unlike prior MLLM speaker work [1, 2] with limited performance (See table 9), we are the first to introduce an MLLM-based instruction generator that outperforms no-MLLM generators and can be further improved through self-training, marking a major contribution to the VLN speaker research.
>
> [1] Navigation Instruction Generation with BEV Perception and Large Language Models, Fan et al. ECCV2024
>
> [2] Controllable Navigation Instruction Generation with Chain of Thought Prompting, Kong et al. ECCV2024
>
> ----
>
> >**Table 5 input/encoding clarification. (R4Q1)**
>
> During training, each combination of one instruction and one trajectory is treated as an individual instance (e.g., three instructions for one trajectory are treated as three separate instances). We do not use multiple instruction encoders to extract additional information beyond a single instruction. As an example, for N paths with k instructions per path, the training dataset will contain N*k instances, with only one instruction encoded by the encoder per instance in the training stage.
>
> ----
>
> >**Table 7 instructions per path. (R4Q2)**
>
> We refer to row 269, where we state that six instructions are generated per path, as shown to be the optimal choice based on the results in Table 5.
>
>
> ----
>
> >**Testing on additional environments. (R4Q3)**
>
> We thank R4 for raising the question about testing on other environments. We selected MP3D-style environments due to their photo-realistic nature, which has been shown to effectively facilitate sim-to-real transfer for real-world robotic VLN [1,2].
>
> Additionally, the primary difference between ALFRED and MP3D environments lies in the visual domain gap—ALFRED uses synthetic images, while MP3D provides photo-realistic scenes. Since our focus is not on addressing this gap, we did not evaluate ALFRED.
>
> Meanwhile, we appreciate R4’s suggestion to explore generalization to out-of-distribution environments and to incorporate abundant synthetic environments (e.g., ProcTHOR) for training. We’ll leave these exciting directions for future work.
>
> [1] Sim-to-Real Transfer for Vision-and-Language Navigation, Anderson et al.
>
> [2] NaVid: Video-based VLM Plans the Next Step for Vision-and-Language Navigation, Zhang et al.
>
>
> ----
>
> We thank again for the valuable feedback and thoughtful discussion. We will incorporate all the above discussions, the mentioned papers, and the results of new experiments into the main paper.

---

> > ### Comment · Reviewer_JHxr · 2024-11-27
> >
> > Thanks for the response.

---

> > > ### Author Response · Authors · 2024-11-27
> > > **Thanks for the response**
> > >
> > > We sincerely thanks for your valuable feedback, thoughtful discussion, and acknowledgment of our work.

---

### Official Review · Reviewer_sXMN · 2024-11-04

**Soundness:** 4
**Presentation:** 3
**Contribution:** 3
**Rating:** 8
**Confidence:** 4

**Summary:**

This paper proposes an iterative instruction-generation and navigation pipeline to increase the quality of machine-labeled data for Vision Language Navigation tasks.  By using a navigation model to identify which generated instructions are achievable, the navigation model can improve the instruction generator's data.  This in turn creates a better instruction generator, which can expand the set of data used to train the navigation model.  The paper shows that by repeating this filtering/training cycle, the resulting navigator can perform comparably (or better than) humans on some benchmarks, and also that the instructions generated by the model are better than other baselines.

**Strengths:**

- The presented method generates impressive empirical results in the navigation task and in the instruction generation task as well.  The evaluations show the importance of carefully selecting (or generating) data to train VLN models.  Although this is not a new observation generally, the thorough and careful application of filtering and data labeling to the VLN problem is a solid contribution with state of the art results.
- Comparison to related work is thorough, both in evaluations and in the text.
- Presentation is clear and highlights strong results without grand claims.  The diagrams are helpful in understanding the approach.  Detailed description of implementation.

**Weaknesses:**

I found few substantial weaknesses in this paper.  Here are some minor suggestions:
- Radar plot in Fig 1. is difficult to read, the Human results are difficult to distinguish from ScaleVLN.
- Line 234 says "navigator $G_1$" which I believe should be $N_1$

**Questions:**

- The paper mentions several times that the navigator model being trained is already very good.  What happens in the case where the navigator is not as good?  Do you expect several more rounds of the flywheel to be required?
- Why does the paper stop at 3 rounds for evaluation?  I expect diminishing returns, but is that a reflection of the lack of high-quality data remaining for the model to label, or something else?
- When discussing label diversity, the paper mentions that the vocabulary size of existing datasets is small.  Can you say more about the effect of vocabulary size?  If, for example, the dataset were relabeled by asking a LLM to rephrase the instructions, would that be sufficient to get better performance?

---

> ### Author Response · Authors · 2024-11-21
> **Response to Reviewer 3 (sXMN)**
>
> We thank Reviewer 3 (sXMN) for acknowledging the contribution of our paper and for the thoughtful comments. Please see our response to the comments below.
>
>
> Regarding the radar plot in Figure 1, we will update it to enhance readability and clearly distinguish the Human results from ScaleVLN. Additionally, we appreciate the detailed feedback on Line 234 and will correct the incomplete phrase to improve clarity in the revised version.
>
> ----
>
> >**Impact of navigator quality on flywheel effectiveness. (R3Q1)**
>
> We thank R3 for the thoughtful questions. In the R2R context, we believe model capacity and scalability are crucial. For example, Seq2Seq [1], with its simplistic LSTM architecture and limited scale, may struggle to benefit largely from the flywheel despite more data.
>
> Instead, scalable models like HAMT [2] and RecBERT [3] leveraging transformers and well-designed state modules, will likely benefit more from our flywheel. HAMT tested under Table 4 settings, achieved 24.7 SPICE and 51.2 CIDEr, showing its effectiveness as a filter. However, it may require more rounds to reach optimal performance as the initial filter is less effective than DUET.
>
> [1] Vision-and-Language Navigation: Interpreting visually-grounded navigation instructions in real environments, Anderson et al.
>
> [2] History Aware Multimodal Transformer for Vision-and-Language Navigation, Chen et al.
>
> [3] A Recurrent Vision-and-Language BERT for Navigation, Hong et al.
>
> ----
>
> >**Reason for 3 rounds. (R3Q2)**
>
> We thank R3 for raising this question. We present three rounds of experiments due to their reasonable scale and affordable computational cost to validate our essential idea of iterative refinement. However, we agree that it is valuable to advance to further rounds of experiments. As shown in the table below, the fourth-round training of the instruction generator using third-round filtered data, achieves even better results of 26.2 SPICE, indicating the strong potential of our method to improve performance with further rounds.
>
> | Method           | SPICE↑ | Bleu-1↑ | Bleu-4↑ | CIDEr↑ | Meteor↑ | Rouge↑ |
> |------------------|--------|---------|---------|--------|---------|--------|
> | Baseline         | 21.8      | 72.5    | 27.7    | 42.2   | 23.6    | 49.0   |
> | Ours (round 1)   | 23.7  | 71.4  | 29.5    | 46.5   | 23.1    | 50.2   |
> | Ours (round 2)   | 25.2  | 73.7    | 31.0    | **50.7**   | 24.2    | 51.3   |
> | Ours (round 3)   | 25.7     | 74.5    | 30.8    | 49.7   | 24.5    | 51.3   |
> | Ours (round 4)   | **26.2**    | **75.3**    | **31.1**    | 49.2   | **25.0**    | **51.4**   |
>
> We also agree with R3 that the lack of remaining low-quality data to refine is a potential bottleneck. We note that by round 3, nearly 90% of the generated instruction can be successfully followed (SR=1), compared to 75% in round 1. This suggests that after a few more rounds, the navigator may saturate, limiting further learning from the data pool.
>
> However, Table 6 demonstrates the scalability of our method, showing more environments lead to improved instruction generation quality. While we currently use MP3D and HM3D environments, we believe expanding to additional environments like Gibson (as in ScaleVLN [4]) could further enhance performance. We plan to explore this direction in future work.
>
> [4] Scaling data generation in vision-and-language navigation, Wang et al.
>
> ----
>
> >**Discussion on vocabulary size. (R3Q3)**
>
> We believe sufficient vocabulary size is crucial for improving grounded text generalization in VLN models, which often struggle due to limited data and vocabulary. In R2R, 50% of the vocabulary in val unseen split is absent in the train split, highlighting the challenge of understanding rare grounded texts. Thus, diverse instructions and large vocabularies are essential, and our approach effectively achieves this, enhancing generalization to rare samples.
>
> Regarding R3’s suggestion to use LLMs for rephrasing R2R instructions, we’ve conducted very similar experiments before but observed no improvement. This was mainly due to hallucination and distribution shifts, as in-context rephrased instructions were not grounded in trajectories and lacked similar instructional styles. We think improving grounded text generalization remains a promising future direction.
>
> ----
> We thanks again for the valuable feedback and thoughtful discussion. We will incorporate all the above discussions, the mentioned papers, and the results of new experiments into the main paper.

---

### Official Review · Reviewer_H4hM · 2024-11-04

**Soundness:** 2
**Presentation:** 3
**Contribution:** 1
**Rating:** 5
**Confidence:** 4

**Summary:**

This paper proposes a framework in which the instruction model and navigation model bootstrap each other through an iterative process. It begins with the pre-trained instruction model labeling sampled trajectories, which are then used to fine-tune the navigator. The updated navigator subsequently samples additional trajectories to further refine the instruction model, and the cycle continues. The authors conduct experiments on various VLN benchmarks, including R2R, CVDN, REVERIE, SOON, R4R, and R2R-CE. The experiment results show that the proposed method is able to

**Strengths:**

1. The paper is well written and easy to follow.
2. The authors have conducted experiments on diverse downstream VLN tasks to validate the effectiveness of the method.

**Weaknesses:**

1. My major concern is the novelty of this paper. The proposed method highly resembles the Speaker Follower model (Fried et al., 2018), and the self-training paradigm has been widely studied in previous studies [1], [2].
2. It’s unclear whether the performance improvements primarily result from the bootstrapping process between the instruction model and navigation model, or from the advantage of initializing the instructor with a pre-trained MLLM (Mantis) that has been trained on a large-scale dataset. From Table 3 we can see that the model improves by a visible margin after round 1, but subsequent rounds yield only marginal gains. Prior studies [3] also indicate that generally, the self-training paradigm does not improve model's capabilities on out of distribution cases by much. It will be interesting to see how model's generalizability improved without relying on the pre-trained MLLM.
3. Some of the notations are not defined clearly. For example, what is D^N_1 and what is the data source of it? It is also unclear to me what is the exact dataset use to produce D_seed.
4. The author mentioned that they use DUET as the baseline navigator, could the author explain the performance discrepancy between the baseline in Table 3 (SR=78) and the DUET model in Table 7 (SR=72)?

In short, I think my major concern is the novelty of the paper. Also, I'm not sure whether the model is benefited by the powerful pre-trained MLLM or the bootstrapping process itself.

[1] Reinforced Self-Training (ReST) for Language Modeling, Gulcehre et al.

[2] Re-ReST: Reflection-Reinforced Self-Training for Language Agents, Dou et al.

[3] The Entropy Enigma: Success and Failure of Entropy Minimization, Press et al.

**Questions:**

1. Why is the performance of the speaker-follower model much lower than the results presented in the original paper?
2. Which dataset and split is used in the experiment reported in Table 3?

---

> ### Author Response · Authors · 2024-11-21
> **Response to Reviewer 2 (H4hM) - Part 1**
>
> We thank Reviewer 2 (H4hM) for the time and effort in reviewing our paper and providing very constructive feedback. Please see our response to the comments below.
>
> ----
>
> >**Paper novelty regarding speaker-follower framework. (R2W1)**
>
> We want to point out that the speaker-follower model is a widely used VLN framework for data augmentation, with many previous works contributing distinct novelties within this scope (see Table 9 for related speaker work and Table 2 “Generated” for data generation using speakers). Our approach builds on this framework but introduces a novel iterative process for mutual improvement between navigator and generator to self-improve the VLN data quality, which is recognized by Reviewer sXMN and JHxr.
>
> Additionally, we also contribute significantly to VLN speaker research, including a strong intuition (Section 3.2) to use what kind of MLLM and effective input strategies (Figure 3, Table 10)  to build a strong generator. Unlike prior MLLM speaker work [1, 2] with limited performance (See table 9), we are the first to introduce an MLLM-based generator that outperforms non-MLLM ones (Table 9) and improves further with self-training, marking a major step in this direction.
>
> [1] Navigation Instruction Generation with BEV Perception and Large Language Models, Fan et al. ECCV2024
>
> [2] Controllable Navigation Instruction Generation with Chain of Thought Prompting, Kong et al. ECCV2024
>
> ----
>
> >**Paper novelty regarding self-improving language model. (R2W1)**
>
> We would like to emphasize that the focus of this paper is **not to develop a general method** for language model self-improvement but to **address the long-lasting data quality challenges** in vision-and-language navigation as shown in the title. To this end, we propose a novel and effective data evaluation and refinement approach—the self-refining data flywheel—to tackle this issue, acknowledged by Reviewer sXMN and JHxr.
>
> The key distinction of our method compared to previous self-improving Language models (LMs) lies in its human-free, two-model mutual improvement process. In contrast to mentioned prior approaches such as [3,4], where the reflection or reward model is fixed to improve only the LM, or RLHF [5], which demands extensive human labeling to collect preference data, our method allows the navigator (considered as a reward model) and language model generator to iteratively improve each other through mutual feedback, without the need for human annotation.
>
> Additionally, We also show that classical self-improvement methods fail in VLN without navigator feedback. Specifically, we test self-rewarding (LM self-score) and CLIP-scoring (CLIP Image-Text similarity) in round 1, filtering 300K instructions for round 2 generator training. Neither improves over the round 1 baseline, highlighting the effectiveness of navigator filtering.
>
> | round          | scorer | SPICE↑ | Bleu-1↑ | Bleu-4↑ | CIDEr↑ | Meteor↑ | Rouge↑ |
> |------------|------|--------|---------|---------|--------|---------|--------|
> |  round 1  | - | 23.7  | 71.4  | 29.5    | 46.5   | 23.1    | 50.2   |
> | round 2  | Self-score  | 23.6     | 71.3    | 29.4    | 46.4   | 23.5    | 50.3   |
> | round 2 | CLIP-score  | 23.9    | 70.6    |  30.0   | 48.6   | 23.1    | 50.4   |
> | round 2 | navigator-nDTW  | **25.2**  | **73.7**    | **31.0**    | **50.7**   | **24.2**    | **51.3**  |
>
> [3] Reinforced Self-Training (ReST) for Language Modeling, Gulcehre et al.
>
> [4] Re-ReST: Reflection-Reinforced Self-Training for Language Agents, Dou et al.
>
> [5] Training language models to follow instructions with human feedback, Ouyang et al.

---

> ### Author Response · Authors · 2024-11-21
> **Response to Reviewer 2 (H4hM) - Part 2**
>
> >**Effect of MLLMs. (R2W2)**
>
> We emphasize that introducing a powerful MLLM does not diminish the contribution of our data flywheel. Adequate model capacity is essential to support the flywheel's ability to generate high-quality instructions and enable continuous refinement. Building the flywheel on a weak instruction generator would be inefficient and unreasonable.
>
> Furthermore, the navigator's improvement in each round is significant, as evidenced by recent VLN literature [2-5]. More importantly, the models refined over 3 rounds (Table 3 final round model) produce the data necessary for training our final model, achieving significantly better performance (Table 7). These results strongly validate the effectiveness of our proposed data flywheel.
>
> We also thank R2 for raising thoughts on Entropy Minimization (EM) [1]. In VLN, evaluation sets are typically assumed to share the same distribution as the training set, so the OOD-related findings in [1] do not directly apply to our results. Additionally, usually EM is a simple training objective for optimizing models with unlabeled data which is unrelated to any flywheel process, making it less relevant and referable to our approach.
>
> To show our model-boosting process is not dependent on Mantis, we test a weaker MLLM, mPLUG-Owl-7B (proposed very early, April 2023) using the same methodology. Despite completing 1.5 rounds due to time constraints, the round 2 generator shows clear improvement with navigator-filtered data, reinforcing the navigator’s role in enhancing the generator and supporting our claim.
> | Method           | NE↓  | OSR↑  | SR↑   | SPL↑  | SPICE↑ |  Bleu-1↑ | Bleu-4↑ | CIDEr↑ | Meteor↑ | Rouge↑ |
> |------------------|-------|-------|-------|-------|--------|----------|---------|---------|--------|---------|
> | Baseline         | 2.37  | 85.5  | 78.6  | 69.9  | 21.8      | 72.5    | 27.7    | 42.2   | 23.6    | 49.0   |
> | DUET + mPLUG-owl (round 1)   | 2.00  | 86.3  | 81.9  | 74.3  | 22.7      | 70.3    | 28.0    | 44.4   | 23.0    | 49.1   |
> | DUET + mPLUG-owl (round 2)  | -  | -  | -  | - | **24.3**    | **72.2**    | **29.1**    | **45.2**   | **23.7**    | **50.0*   |
>
>
>
> [1] The Entropy Enigma: Success and Failure of Entropy Minimization, Press et al.
>
> [2] Volumetric Environment Representation for Vision-Language Navigation, Liu et al. CVPR2024
>
> [3] PRET: Planning with Directed Fidelity Trajectory for Vision-and-Language Navigation, Lu et al. ECCV2024
>
> [4] NavGPT-2: Unleashing Navigational Reasoning Capability for Large Vision-Language Models, Zhou et al. ECCV2024
>
> [5] Vision-Language Navigation with Energy-Based Policy, Liu et al. NIPS2024

---

> ### Author Response · Authors · 2024-11-21
> **Response to Reviewer 2 (H4hM) - Part 3**
>
> >**Notations clarification. (R2W3)**
>
> We thank R2 for pointing out the notation issue. We acknowledge that while the notation D^N_1 is referenced elsewhere, such as in Figure 2 and row 244, it was missed in row 228, where it refers to the generated data used to train the navigator in round 1. We will address this in the revised version.
>
> Regarding D_seed, it is already defined in row 266 as the R2R dataset. We hope this resolves any confusion about the notation.
>
> ----
>
> >**DUET performance difference between Table 3 and 7. (R2W4)**
>
> We would like to point out that DUET is a VLN model architecture. The original DUET (Table 7) used Prevalent for augmentation, while the baseline DUET (Table 3) used both ScaleVLN-HM3D and Prevalent, improving SR from 72 to 78. We adopt this 78-SR baseline for a fair comparison, as our data generation is built upon ScaleVLN-HM3D and Prevalent environments.
>
> ----
>
> >**Speaker-follower performance difference. (R2Q1)**
>
> We note that the best performance reported in the Speaker-Follower paper is based on the beam-search setting, which sacrifices efficiency to improve success rates by exploring multiple candidate viewpoints at each step and selecting the trajectory with the highest score. This setting is impractical for real-world robot navigation. Table 7 reports Speaker-Follower results in the practical and standard single-run setting.
>
> ----
>
> >**Evaluation dataset for Table 3. (R2Q2)**
>
> We thank R2 for pointing out the missing dataset splits in some tables. We clarify that the results in Tables 3, 4, and 5 are evaluated on the val_unseen split, and we will update the paper to make the evaluation split clearer.
>
> ----
> We thanks again for the valuable feedback and thoughtful discussion. We will incorporate all the above discussions, the mentioned papers, and the results of new experiments into the main paper.

---

> > ### Author Response · Authors · 2024-11-25
> > **A Gentle Reminder**
> >
> > Dear Reviewer H4hM,
> >
> > Thank you for your time and thoughtful feedback on our paper. As the discussion period nears its conclusion, we kindly ask if our responses have addressed your concerns. Specifically, during the rebuttal phase, we have:
> >
> > - Discussed our scope and novelty regarding VLN, speaker-follower, and self-improving framework research, highlighting challenges in directly applying existing self-improving methods to VLN with new results.
> > - Provided detailed discussions and new results regarding effect of MLLM along with per-round improvement clarification.
> > - Revised our paper to address missing notations and evaluation dataset split (line 324, 378-379, 227-228).
> > - Added all additional experiments + discussions (lines 994-1050), alongside revisions to the “self-improving language model” related work (lines 154-161) to better emphasize our unique contributions.
> >
> > We hope the clarifications and updates sufficiently address your concerns, and we kindly ask you to consider these revisions in your evaluation. Please let us know if further information or results are needed.
> >
> > Best,
> >
> > Authors

---

> > > ### Comment · Reviewer_H4hM · 2024-11-27
> > > **Thanks for the response**
> > >
> > > Thank you for the detailed response! I still have a couple of follow-up questions:
> > >
> > > 1. The authors highlight that "the key distinction of our method compared to previous self-improving language models (LMs) lies in its human-free, two-model mutual improvement process." However, this distinction might benefit from further clarification. The cited related works [3,4] also employ a human-free approach, leveraging environmental feedback (e.g., success or failure) to facilitate model improvement, which appears conceptually similar. Additionally, these works refine both the prediction model (analogous to the navigator in this context) and the reward model (similar to the speaker here) using the collected trajectories. While the authors emphasize their focus on challenges specific to the VLN community rather than proposing a general-purpose learning method, the contribution could be framed more clearly to demonstrate how the application of self-training/self-reflection in VLN provides unique insights or advances the field.
> > >
> > > 2. As reviewer FkSF has pointed out, the manuscript does not address the theoretical or practical limitations of applying self-improvements to VLN tasks. A deeper exploration of these aspects could strengthen the paper by positioning the framework more clearly within the broader landscape of self-improving models. For example, discussing how the mutual improvement process might overcome specific challenges in VLN or outlining its potential boundaries would provide valuable context and enhance the novelty of the work.

---

> ### Author Response · Authors · 2024-11-27
> **2nd Response to Reviewer 2 (H4hM) - Part (1/3)**
>
> We appreciate your detailed response and thoughtful questions! Please see our response to the comments below.
>
> ----
>
> >**Comparison to [3,4] and analogy to the self-training framework**
>
> We would like to point out that a self-improving language model (LM) typically refers to a model that iteratively improves using its own generated data. In our framework, **this corresponds to the speaker, not the navigator**, as the speaker is refined based on its self-generated instructions (we also mentioned this in Related Work lines 159-160). The navigator, providing feedback to the speaker's data for filtering, acts as a reward model (RM). In this regard, similar to prior RLHF/self-improving LM methods, we show that RM improves LM.
>
> On the other hand, we also demonstrate that **LM improves RM** in the VLN context, which is distinct from most previous works, including [3,4]. In our method, the RM navigator is refined by the LM generator through data-pool refinement, establishing a two-model mutual improving process where the RM navigator and LM generator iteratively enhance each other.
>
> Additionally, to comprehensively compare [3], [4] with our method, we build the table below with detailed references from their main papers.
>
> As shown in the table, neither [3] nor [4] demonstrate a mutual improving process:
>  - both the RMs are **fixed** after initial training
> - both methods focus **solely on improving LM with RM**
> - [3] relies on **human-collected preference data** to train the initial RM, making it not fully human-free
> - [4] only conducts **one-round** LM-improvement experiments
>
> In contrast, we demonstrate a two-model mutual improvement process: not only does the RM navigator enhance the LM generator, but the LM generator also consistently improves the RM navigator through multi-round flywheel iterations, entirely human-free.
>
> We hope this resolves any confusion about the distinctiveness of our method.
>
> | **Feature**                          | **ReST [3]** | **Re-ReST [4]** | **Ours** | **Reference in [3]**                                                                                                                                              | **Reference in [4]**                                                                                                     |
> |--------------------------------------|--------------|-----------------|-----------|-------------------------------------------------------------------------------------------------------------------------------------------------------------------|--------------------------------------------------------------------------------------------------------------------------|
> | Reward Model Improving Language Model | yes          | yes             | yes       | -                                                                                                                                                                 | -                                                                                                                        |
> | Human-Free                          | no           | yes             | yes       | Introduction, 3rd paragraph --  "We use a learned reward model trained on human preferences as the scoring function" -- so require humans to annotate preference data first | -                                                                                                                        |
> | Language Model Improving Reward Model | no           | no              | yes       | Algorithm 1 -- The reward model is only a fixed required input, and the algorithm is used to train only the LM Pi_theta -- so the reward model is not improved at all | Section 2.3, 1st paragraph -- Pipeline: Train reflection model R -> R generates data D_R + Language Model M generates data D_M -> D_M + D_R train M -- so it’s still reflection model improving LM, but no LM improving RM involved |
> | Multi-Round Improving                         | yes          | no              | yes       | -                                                                                                                                                                 | Still Section 2.3, 1st paragraph -- Pipeline: Train reflection model R -> R generates data D_R + Language Model M generates data D_M -> D_M + D_R train M --  so there’s only one round of LM improving                                                                                     |
>
> [3] Reinforced Self-Training (ReST) for Language Modeling, Gulcehre et al.
>
> [4] Re-ReST: Reflection-Reinforced Self-Training for Language Agents, Dou et al.

---

> ### Author Response · Authors · 2024-11-27
> **2nd Response to Reviewer 2 (H4hM) - Part (2/3)**
>
> > **How mutual improving process solves VLN challenges**
>
> Thanks for your thoughtful suggestion. As discussed, our work highly addresses the **data quality challenge** in VLN. In previous speaker-follower research, limited high-quality data (e.g., only 14K human-labeled examples in R2R) led to poor generalization of the generator, making it difficult to produce enough high-fidelity data for training a stronger generator (+ navigator). Our mutual improving process addresses this: the navigator filtering process provides additional high-quality data (beyond the 14K samples) to train a stronger generator,  and the improved generator then produces better data to enhance the navigator. When running the process iteratively, the ever-improving generator builds a higher-quality data pool, while the ever-better navigator provides increasingly reliable feedback, ultimately producing a substantially high-quality VLN dataset (along with both a very strong generator and navigator), effectively addressing the data quality issue in VLN.
>
> Additionally, It's challenging for the navigator or generator to self-evolve:
> - Navigator: The navigator relies on generator-produced data and, without data refining by an improved generator, it is trained on the same dataset repeatedly. Moreover, as shown in Table 5 of the main paper, even doubling the data pool (#Instr=12) does not improve the navigator’s performance, highlighting the necessity of refining the data pool with an improved generator rather than merely scaling it.
>
> - Generator: As noted in our response (Reviewer 2 (H4hM) - Part 1, "Paper novelty regarding self-improving language model (R2W1)"), the generator’s self-score failed to improve itself, while the navigator's nDTW improved the generator significantly, emphasizing the critical role of navigator feedback in enhancing the generator.
>
> These highlight the importance of our mutual improving process to improve models and data simultaneously.

---

> > ### Author Response · Authors · 2024-11-27
> > **2nd Response to Reviewer 2 (H4hM) - Part (3/3)**
> >
> > >**Discussions on potential performance boundaries**
> >
> > In our main paper, we conducted 3-round experiments to validate our main idea, but the performance doesn’t saturate within three rounds: As shown in the table below, the fourth-round training of the instruction generator using third-round navigator-filtered data, achieves even better results of 26.2 SPICE, indicating the strong potential of our method to improve performance with further rounds.
> >
> > | Method           | SPICE↑ | Bleu-1↑ | Bleu-4↑ | CIDEr↑ | Meteor↑ | Rouge↑ |
> > |------------------|--------|---------|---------|--------|---------|--------|
> > | Baseline         | 21.8      | 72.5    | 27.7    | 42.2   | 23.6    | 49.0   |
> > | Ours (round 1)   | 23.7  | 71.4  | 29.5    | 46.5   | 23.1    | 50.2   |
> > | Ours (round 2)   | 25.2  | 73.7    | 31.0    | **50.7**   | 24.2    | 51.3   |
> > | Ours (round 3)   | 25.7     | 74.5    | 30.8    | 49.7   | 24.5    | 51.3   |
> > | Ours (round 4)   | **26.2**    | **75.3**    | **31.1**    | 49.2   | **25.0**    | **51.4**   |
> >
> > We believe a potential bottleneck lies in the diminishing availability of low-quality data for further refinement due to fixed trajectory pool in MP3D+HM3D environments. As noted in our response to Reviewer 3 (sXMN) R3Q2, by round 3, nearly 90% of generated instructions are successfully followed (SR=1), compared to 75% in round 1. This suggests that after a few additional rounds, navigator performance may saturate, as the generator-produced instructions will all be followable, limiting further learning opportunities from the data pool.
> >
> > However, Table 6 demonstrates the scalability of our method, showing more environments lead to improved instruction generation quality. While we currently use MP3D and HM3D environments, we believe expanding to additional environments like Gibson (as in ScaleVLN [1]), ProcThor (mentioned by Reviewer JHxr), etc, could further enhance performance. We plan to explore this direction in future work.
> >
> > [1] Scaling data generation in vision-and-language navigation, Wang et al.

---

> > > ### Author Response · Authors · 2024-11-29
> > > **The end of the discussion stage is coming**
> > >
> > > Dear Reviewer H4hM,
> > >
> > > Thank you for your thoughtful feedback on our paper. As the discussion period comes to a close, we kindly ask you to review our latest responses. We hope the clarifications and updates sufficiently address your concerns, and we would greatly appreciate it if you could consider these updates in your evaluation. Please let us know if any additional information is needed.
> > >
> > > Thank you again for your valuable input and time.
> > >
> > > Best,
> > >
> > > Authors

---

> > > > ### Comment · Reviewer_H4hM · 2024-12-01
> > > > **Thanks for the response**
> > > >
> > > > I would like to thank the authors for their detailed response.
> > > >
> > > > However, I still have concerns regarding the technical novelty of the paper.
> > > >
> > > > After checking [4], I would like to respectfully disagree with the author's claim: "it’s still reflection model improving LM, but no LM improving RM involved". In [4], the reflection model (analogous to the reward model, RM) is trained on a dataset generated by the language agent (analogous to the navigator LM), while the language agent benefits from the additional trajectories collected through the reflection process. A similar technique is also proposed in the well-known Constitutional AI paper [5], where data sampled from an LM is used to train the RM, which then provides feedback to fine-tune the LM. As a result, I find the relationship between the LM and RM in the proposed work appears to align closely with existing methods.
> > > >
> > > > [5] Constitutional AI: Harmlessness from AI Feedback. Bai et al.

---

> ### Author Response · Authors · 2024-12-02
> **3rd Response to Reviewer 2 (H4hM) - Part (1/2)**
>
> Thanks for providing additional missing related work and the detailed response. Please see our response below.
>
> > **Method-level comparison with self-improving/RLHF/RLAIF**
>
> Our mutual improving process has three key attributes:
> - (1) Two models can improve each other
> - (2) the first process can run multiple rounds
> - (3) entirely human-free.
>
> Regarding the mentioned papers by the reviewer:
>
> We agree that the reflection model in [4] is initially trained on LM-generated data, so it may satisfy (1). However, as we discussed in our comparison table, [4] only demonstrates a **single-round** improvement, **failing to meet (2)**. Additionally, a key difference in [4] is, that both models act as data generators, unlike our framework (or RLHF framework) with one generator (LM) and one discriminator (RM).
>
> We also thank the reviewer for highlighting the missing papers in the RLAIF field [5-11]; we will include them in our related work. However, we’d like to point out that Constitutional AI [5] still relies on human labeling — as noted in Section 1.2, 3rd paragraph (RL Stage), “***we distill LM interpretations of a set of principles back into a hybrid human/AI PM (using human labels for helpfulness, but only AI labels for harmlessness)***” — so it still relies on human in the helpfulness side, **not entirely human-free, which doesn’t satisfy (3)**. Additionally, we don’t see a multi-round iterative improvement process between the LM and RM, as the RM (referred to as the preference model in the paper) is only trained once, so it **may also fail to meet (2)**.
>
> Moreover, we found lots of related RLAIF papers [6-11] use **fixed foundation VLMs/LLMs/RMs [6-11] to provide feedback, making them doesn’t satisfy (1)**. We also tested initial results with fixed VLM feedback in our case (See “Response to Reviewer 2 (H4hM) - Part 1, Paper novelty regarding self-improving language model. (R2W1)”, table is also copied below), showing that CLIP feedback failed to improve the LM generator. Combined with the failure of self-score, **we show that directly applying classical self-rewarding/RLAIF methods fails in the VLN context**, highlighting VLN challenges and the importance of navigator feedback.
>
> We hope these discussions clarify the distinctiveness of our methods (proposed for VLN).
>
> | round          | scorer | SPICE↑ | Bleu-1↑ | Bleu-4↑ | CIDEr↑ | Meteor↑ | Rouge↑ |
> |------------|------|--------|---------|---------|--------|---------|--------|
> |  round 1  | - | 23.7  | 71.4  | 29.5    | 46.5   | 23.1    | 50.2   |
> | round 2  | Self-score  | 23.6     | 71.3    | 29.4    | 46.4   | 23.5    | 50.3   |
> | round 2 | CLIP-score  | 23.9    | 70.6    |  30.0   | 48.6   | 23.1    | 50.4   |
> | round 2 | navigator-nDTW  | **25.2**  | **73.7**    | **31.0**    | **50.7**   | **24.2**    | **51.3**  |
>
>
>
> [4] Re-ReST: Reflection-Reinforced Self-Training for Language Agents, Dou et al.
>
> [5] Constitutional AI: Harmlessness from AI Feedback, Bai et al.
>
> [6] RLAIF vs. RLHF: Scaling Reinforcement Learning from Human Feedback with AI Feedback, Lee et al. ICML2024
>
> [7] RL-VLM-F: Reinforcement Learning from Vision Language Foundation Model Feedback, Wang et al. ICML2024
>
> [8] Math-Shepherd: Verify and Reinforce LLMs Step-by-step without Human Annotations, Wang et al. ACL2024
>
> [9] Accelerating Reinforcement Learning of Robotic Manipulations via Feedback from Large Language Models, Chu et al. CoRL2024 workshop
>
> [10] RLAIF-V: Aligning MLLMs through Open-Source AI Feedback for Super GPT-4V Trustworthiness, Yu et al. ArXiv2024
>
> [11] Enhancing Robotic Manipulation with AI Feedback from Multimodal Large Language Models, Liu et al. AAAI2024 workshop

---

> ### Author Response · Authors · 2024-12-02
> **3rd Response to Reviewer 2 (H4hM) - Part (2/2)**
>
> > **Novelty Discussions**
>
> While high-level conceptual similarities across fields can be noted, we believe **they may not necessarily overshadow the novelty within a specific field** (especially considering large field differences like differences between VLN and self-improving LMs). For example, the reviewer-mentioned speaker-follower framework [12]  in VLN (2018 proposed) shares similarities with the back-translation method [13] in NMT (2016 proposed), yet its contribution remains significant in VLN. Similarly, we also notice that [6-11] contribute by applying RLAIF to problems including manipulation, math, control, summarization, dialogue generation, trustworthiness, etc. while their method on a high-level may be conceptually similar to the first RLAIF paper, Constitutional AI [5].
>
> We also emphasize that our technical novelty lies in how we address the data quality issue in VLN, which involves two key aspects: evaluating and improving the data.
> - For evaluation, we propose using the navigator for self-evaluation, which has not been proposed in VLN before, and *we prove it to be more effective than classical VLM scores.
> - For improving data quality, we propose to iteratively refine the data pool with an ever-improved generator trained with ever-improved-navigator-filtered data, through running our self-refining data flywheel. Such a two-model mutual improving process is also novel in VLN, as previous work only involves one generator-augmentation step. Importantly, our approach is highly effective, demonstrated by the creation of a high-quality synthetic VLN dataset, which leads to **significantly stronger navigator results across seven VLN tasks over SoTA, even outperforming or approaching human performance for the first time in some cases.**
>
> On the other hand, as noted earlier, we demonstrate that classical self-rewarding or AI-feedback methods do not improve generator performance, underscoring the unique challenges in VLN and the critical role of navigator feedback. We hope this clarifies the novelty of our approach within the VLN domain.
>
> [12] Speaker-Follower Models for Vision-and-Language Navigation, Fried et al. NIPS2018
>
> [13] Improving Neural Machine Translation Models with Monolingual Data, Sennrich et al. ACL2016
>
> ----
>
> > **Potential new insights of our method to self-improving/RLHF/RLAIF fields**
>
> We agree with the reviewer that the **cross-field inspiring thoughts** are more valuable (mentioned by the reviewer: “*the contribution could be framed more clearly to demonstrate how the application of self-training/self-reflection in VLN provides unique insights or advances the field*”) rather than focusing on arguing conceptual similarities using high-level analogies, and thus we highlight the potential new insights of our method to self-improving/RLHF/RLAIF areas.
>
> Our work on VLN may offer new insights into the general self-improving field: **With the advancement of strong LMs, the RM also has the potential to be iteratively improved using LM-generated data, and establish a multi-round mutual improving process between the LM and the RM**, which can be potentially applied to building general language agents, or some specific fields like  (embodied) vision-and-language, etc (considering VLN is a challenging embodied vision-and-language task metioned in [14]).
>
> And, as seen with our RM navigator being iteratively improved and **finally outperforming or approaching humans on several VLN tasks**, iteratively improving RM could potentially lead to **AI-trained RM achieving (nearly) human-level performance, reducing or replacing the human role in RLHF** and making RLAIF generally more competitive with RLHF.
> We consider this a potential direction for future research.
>
>
> [14] Retrospectives on the Embodied AI Workshop, Deitke et al. ArXiv2022
>
> ----
>
> We thanks again for the valuable feedback and thoughtful discussion. We will incorporate all the above discussions, the mentioned papers, and the results of new experiments into the main paper.

---

### Official Review · Reviewer_FkSF · 2024-11-12

**Soundness:** 3
**Presentation:** 3
**Contribution:** 2
**Rating:** 5
**Confidence:** 3

**Summary:**

This paper introduces the Self-Refining Data Flywheel pipeline within the context of Vision-and-Language Navigation (VLN). This pipeline iteratively improves itself by generating navigation instructions and refining navigation paths through repeated cycles of self-improvement. The goal is to enhance VLN performance by creating a feedback loop that iteratively optimizes both instruction generation and navigation, facilitating autonomous performance improvements over time.

**Strengths:**

- Simplicity and Effectiveness: The concept of iteratively refining instructions and navigation steps is straightforward, yet the experimental results validate its utility in enhancing VLN task performance.

- Clarity and Structure: The paper is well-written and clear, with a structured presentation that helps readers understand the pipeline. Additionally, it effectively contextualizes the work within existing VLN research, making its contributions easy to follow.

- SoTA performance: While I'm not an expert in the fields of VLN, it seems the paper achieved the good performance compared to the prior SoTA>

**Weaknesses:**

- Terminology and Scope: The term "Flywheel" might be overly strong for what is essentially a straightforward iterative self-improvement process, and it could be perceived as a bit exaggerated.

- Limited Dataset Scope: The method is only tested on the R2R as a base dataset, and the lack of the discussion on its broader applicability.

- Limited Analysis of Performance Boundaries: While performance improvements are reported in up to three rounds (Table 3), little discussion exists on why the improvement plateaus and whether there are theoretical or practical limits to the approach’s performance gains.

- Positioning within Broader Self-Improving Frameworks: The novelty of this approach needs to be clarified when considering related research beyond VLN. The "Self-Improving Language Models" section lacks a strong position on how this framework differentiates itself within the wider landscape of self-improving models.

**Questions:**

1. How does performance change when using a different baseline from R2R? Could the choice of base data impact the outcomes, and if so, how?

2. SPL and nDTW are used as metrics for improvement. Could you explain why these were chosen specifically, and how would the results differ with alternative metrics?

3. Considering research outside of VLN, what novel aspects of this framework make it distinct within the scope of self-improving models?

---

> ### Author Response · Authors · 2024-11-21
> **Response to Reviewer 1 (FkSF) - Part 1**
>
> We thank Reviewer 1 (FkSF) for the time and effort in reviewing our paper and providing very constructive feedback. Please see our response to the comments below.
>
> ----
>
> >**Terminology appropriateness and scope clarification. (R1W1)**
>
> We thank R1 for the question about terminology. A "flywheel" [1] refers to an iterative process where models generate data, which is refined and used to train improved models, creating a self-reinforcing cycle. In our approach, the generator produces data, the navigator provides feedback to refine it, and the refined data improves both the generator and navigator. This process also accommodates introducing additional data, such as new environments with annotation from the language model. Overall, this aligns with the broader concept of a data flywheel.
>
> [1] The flywheel effect, Kanold et al.
>
> ----
> >**Only use R2R as the base dataset. (R1W2)**
>
> We would like to point out that our focus is on improving VLN data quality, not proposing a new LM self-improvement method. R2R is carefully chosen as it combines key attributes of various VLN datasets: fine-grained semantic and directional clues (RxR-en, CVDN) and diverse stopping guidance (REVERIE, SOON). This ensures the broad applicability of the final navigator to various downstream VLN tasks, as evidenced by its strong generalization and effectiveness across seven VLN tasks beyond R2R in Tables 7 and 8 in the paper.
>
> ----
>
> >**Performance plateau and scalability limitations. (R1W3)**
>
> We thank R1 for raising the question. We present three rounds of experiments due to their reasonable scale and affordable computational cost to validate our essential idea of iterative refinement. However, we agree with R1 that it is valuable to advance to further rounds of experiments. As shown in the table below, the fourth-round training of the instruction generator using third-round filtered data, achieves even better results of 26.2 SPICE, indicating the strong potential of our method to improve performance with further rounds.
>
> | Method           | SPICE↑ | Bleu-1↑ | Bleu-4↑ | CIDEr↑ | Meteor↑ | Rouge↑ |
> |------------------|--------|---------|---------|--------|---------|--------|
> | Baseline         | 21.8      | 72.5    | 27.7    | 42.2   | 23.6    | 49.0   |
> | Ours (round 1)   | 23.7  | 71.4  | 29.5    | 46.5   | 23.1    | 50.2   |
> | Ours (round 2)   | 25.2  | 73.7    | 31.0    | **50.7**   | 24.2    | 51.3   |
> | Ours (round 3)   | 25.7     | 74.5    | 30.8    | 49.7   | 24.5    | 51.3   |
> | Ours (round 4)   | **26.2**    | **75.3**    | **31.1**    | 49.2   | **25.0**    | **51.4**   |
>
> Additionally, we refer to paper's Table 6 to highlight the scalability of our method, showing that increasing the number of environments improves instruction generation results. Currently, we build our data flywheel upon MP3D and HM3D environments, but expanding to additional environments, such as Gibson (used in ScaleVLN [2]), is possible to further enhance performance, as suggested by paper's Table 6 and ScaleVLN’s Table 2. We plan to explore this direction in future work.
>
> [2] Scaling data generation in vision-and-language navigation, Wang et al.

---

> > ### Author Response · Authors · 2024-11-21
> > **Response to Reviewer 1 (FkSF) - Part 2**
> >
> > >**Novelty and positioning in self-improving frameworks. (R1W4 + R1Q3)**
> >
> > we would like to emphasize that the focus of this paper is **not to develop a general method** for language model self-improvement but to **address the long-lasting data quality challenges** in vision-and-language navigation as shown in the title. To this end, we propose a novel and effective data evaluation and refinement approach—the self-refining data flywheel—to tackle this issue, verified and suggested by Reviewer sXMN and JHxr.
> >
> > The key distinction of our method compared to previous self-improving Language models (LMs) lies in its human-free, two-model mutual improvement process. In contrast to prior approaches rely solely on LM or involve an LM with a fixed reward model to improve only the LM, our method allows the navigator (considered as a reward model) and the language model generator to iteratively improve each other through mutual feedback, without the need for human annotation.
> >
> > Additionally, we show that classical LM self-improvement methods fail in VLN without reliable navigator feedback. Specifically, we test self-rewarding (LM self-score) and CLIP-scoring (CLIP Image-Text similarity) in round 1, filtering 300K instructions for round 2 generator training. Neither improves over the round 1 baseline, highlighting the effectiveness of navigator feedback for filtering.
> > | round          | scorer | SPICE↑ | Bleu-1↑ | Bleu-4↑ | CIDEr↑ | Meteor↑ | Rouge↑ |
> > |------------|------|--------|---------|---------|--------|---------|--------|
> > |  round 1  | - | 23.7  | 71.4  | 29.5    | 46.5   | 23.1    | 50.2   |
> > | round 2  | Self-score  | 23.6     | 71.3    | 29.4    | 46.4   | 23.5    | 50.3   |
> > | round 2 | CLIP-score  | 23.9    | 70.6    |  30.0   | 48.6   | 23.1    | 50.4   |
> > | round 2 | navigator-nDTW  | **25.2**  | **73.7**    | **31.0**    | **50.7**   | **24.2**    | **51.3**  |

---

> ### Author Response · Authors · 2024-11-21
> **Response to Reviewer 1 (FkSF) - Part 3**
>
> >**Impact of base data on performance. (R1Q1)**
>
> We emphasize that large-scale data generation demands significant resources, making careful base data selection essential. We chose R2R as it combines key attributes of various VLN datasets: fine-grained semantic and directional clues (RxR-en, CVDN) and diverse stopping guidance (REVERIE, SOON). This ensures broad applicability to downstream VLN tasks.
>
> As shown in paper's Tables 7 and 8, using R2R as base data, our flywheel-data-pretrained navigator serves as a "foundation model" for VLN, achieving significant improvements over state-of-the-art methods across all seven tasks after fine-tuning. This demonstrates that vln-task-specific flywheels may be unnecessary, as our pre-trained navigator generalizes effectively, contributing toward a general foundation model for VLN.
>
> ----
>
> >**Choice of filtering metrics and impact of alternatives. (R1Q2)**
>
> We use SPL [2] and nDTW [1] because they are key path-fidelity metrics to evaluate how closely the navigator’s path aligns with the ground truth. Both score 1 for perfect alignment (row 103). Unlike success, which ignores intermediate path quality, SPL and nDTW capture true path alignment, avoiding issues like overly long or unrealistic paths achieving high scores.
>
> We validate this by extending paper's Table 4 below with success and Oracle success as filters, showing limited effectiveness due to noise in intermediate paths. Additionally, Table 4 evaluates classical scoring functions, confirming that nDTW reliably identifies well-aligned instructions, while CLIP and Mantis scores are less reliable, highlighting the importance of path fidelity metrics for evaluating instruction alignment.
>
> | Filter            | SPICE↑ | SPICE-D↑ | CIDEr↑ |
> |-------------------|--------|----------|--------|
> | No Filter         | 23.7   | 28.4     | 46.5   |
> | CLIP-Sim          | 24.4   | 28.7     | 45.8   |
> | Mantis-Score      | 23.6   | 28.2     | 48.3   |
> | Navigator-Oracle Success    | 23.4   | 28.2     | 49.3   |
> | Navigator-Success    | 24.0   | 28.8     | 50.0   |
> | Navigator-nDTW    | **25.4**   | **30.6**     | **53.9**   |
>
> [1] General evaluation for instruction conditioned navigation using dynamic time warping, Ilharco et al.
>
> [2] On evaluation of embodied navigation agents, Anderson et al.
>
> ----
> We thank again for the valuable feedback and thoughtful discussion. We will incorporate all the above discussions, the mentioned papers, and the results of new experiments into the main paper.

---

> > ### Author Response · Authors · 2024-11-25
> > **A Gentle Reminder**
> >
> > Dear Reviewer FkSF,
> >
> > Thank you for your time and thoughtful feedback on our paper. As the paper revision deadline is coming, we kindly ask if our responses have addressed your concerns. Specifically, during the rebuttal phase, we have:
> >
> > - Discussed our scope and novelty regarding VLN and self-improving framework research, highlighting challenges in directly applying existing self-improving methods to VLN with new results.
> > - Included additional fourth-round training results + referred results in the main paper and discussions to demonstrate the extensibility and scalability of our method.
> > - Clarified the motivations (with supporting results in the main paper) behind our base data selection.
> > - Provided detailed reason for our use of the term “flywheel”.
> > - Elaborated on the reason for metric selection and provided new results regarding its effect.
> > - Added all additional experiments + discussions (lines 994-1050), alongside revisions to the “self-improving language model” related work (lines 154-161) to better emphasize our unique contributions.
> >
> > We hope the clarifications and updates sufficiently address your concerns, and we kindly ask you to consider these revisions in your evaluation. Please let us know if further information or results are needed.
> >
> > Best,
> >
> > Authors

---

> ### Author Response · Authors · 2024-11-29
> **The end of the discussion stage is coming**
>
> Dear Reviewer FkSF,
>
> Thank you for your thoughtful feedback on our paper. As the discussion period comes to a close, we kindly ask you to review our latest responses. We hope the clarifications and updates sufficiently address your concerns, and we would greatly appreciate it if you could consider these updates in your evaluation. Please let us know if any additional information is needed.
>
> Thank you again for your valuable input and time.
>
> Best,
>
> Authors

---

### Meta-Review · Area_Chair_k9Jr · 2024-12-24

**Metareview:**

This paper presents an iterative framework for vision-and-language navigation (VLN) that improves data quality and model performance through mutual refinement of an instruction generator (speaker) and a navigator. By using the navigator to filter generated data and iteratively updating both models with AI feedback, the pipeline is able to create high-quality synthetic data without requiring human annotations. Experiments across eight VLN benchmarks show strong performance, including surpassing human-level navigation results on R2R and achieving state-of-the-art outcomes on other tasks. The approach focuses on scalability, generalization, and provides a robust solution to the data quality challenges in VLN.

All reviewers appreciated the clear presentation, strong results, and detailed experimental evaluation. However, two reviewers raised concerns regarding (1) the naming of the method and (2) the novelty and differentiation of the method from prior work, particularly with respect to self-improving models. These concerns were partially addressed during the rebuttal phase. All reviewers acknowledged and responded to the rebuttal.

The AC recommends acceptance for this submission, recognizing its significant contributions and alignment with the reviewers’ consensus on the strengths of the method, including its presentation, results, and experimental rigor. While the reviewers had raised a concern about the method name, it seems reasonable within the context of the work. However, providing a brief explanation of the naming rationale in the final version could help address any lingering ambiguity. The AC also suggests that the authors include an upfront discussion of connections to self-improving language models and RLAIF, as authors acknowledge in discussions with reviewers (e.g., '3rd Response to Reviewer 2 (H4hM) - Part (2/2)'). The authors’ responses and clear writing already show a strong effort towards addressing these connections, which, if expanded further in the final version, would situate the work even more effectively within the broader research landscape.

**Additional Comments On Reviewer Discussion:**

All four reviewers appreciated the clear presentation, strong results, and detailed experimental evaluation. No ratings were changed - 8, 8, 5, and 5. All reviewers acknowledged and responded to the rebuttal.

---
---

Two concerns, one from FkSF and another from H4hM (score at 5):

- Reviewer FkSF raised concerns regarding (1) the naming of the method (use of "data flywheel").

AC weightage: The term 'data flywheel' is fairly common now, and it seems reasonable within the context of the work.

- Reviewer H4hM raised concerns about (2) the novelty and differentiation of the method from prior work, particularly with respect to self-improving models.

AC weightage: There were no technical errors highlighted in the review, mostly noted overlap with research in self-improving language models. However, there isn't significant overlap with prior methods in VLN research. These concerns were partially addressed during the rebuttal phase. The authors engaged constructively and improved the description, also including an upfront discussion of connections between their method and self-improving language models as well as RLAIF.

---

### Decision · Program_Chairs · 2025-01-22

Accept (Poster)